# Artificial non-monotonic neurons based on nonvolatile anti-ambipolar transistors

Yue Pang [1], Yaoqiang Zhou [1,2] ✉, Shirong Qiu[1], Lei Tong[1], Ni Zhao [1] & Jian-Bin Xu [1,3] ✉

Non-monotonic neurons integrate monotonic input into a non-monotonic response, effectively improving the efficiency of unsupervised learning and precision of information processing in peripheral sensor systems. However, non-monotonic neuron-synapse circuits based on conventional technology require multiple transistors and complicated layouts. By leveraging the advantages of compact design for complex functions with two-dimensional materials, herein, we used anti-ambipolar transistor with airgaps configuration to fabricate the non-monotonic neuron with a bell-shaped response function. The anti-ambipolar transistor demonstrated near-ideal subthreshold swings of 60 mV/dec, a benchmark combination of a high peak-to-valley ratio of $\sim 10^5$. By utilizing the floating gate architecture, the non-volatile transistors achieved a high operating speed $\sim 10^{-7}$ s and robust durability exceeding $10^4$ cycles. The non-volatile anti-ambipolar transistor showed spike amplitude, width, and number-dependent excitation and inhibition synaptic behaviors. Furthermore, its non-volatile performance can replicate biological neurons showing a reconfigurable monotonic and non-monotonic response by modulating the amplitude and width of presynaptic input. We encoded systolic blood pressure and resting heart rate data to train non-monotonic neurons, achieving the prediction of health conditions with a detection accuracy surpassing 85% at the device level, closely corresponding to the recognized medical standards.

The transformation of monotonic stimuli from sensory inputs into non-monotonic representations is fundamental to how humans process and respond to complex signals[1,2]. At the sensory periphery, neurons typically exhibit monotonic stimulus-response curves, where their activity increases proportionally with stimulus intensity. While this monotonic encoding is effective for initial signal detection, it is insufficient for distinguishing signals of specific intensities that carry critical contextual information. Non-monotonic neurons, also known as intensity-tuned neurons[3], overcome this limitation by selectively responding to intermediate stimulus ranges[4]. This transformation enhances computational efficiency and reduces the complexity of physiological data processing. For instance, systolic blood pressure (SBP) and resting heart rate (RHR)—key indicators of cardiovascular health—often fall within moderate ranges associated with healthy conditions. Effective processing of these signals requires recognizing and responding to intermediate intensity ranges rather than single-side extreme values, making non-monotonic encoding essential. Inspired by this biological mechanism, artificial non-monotonic neurons with dual thresholds can be implemented in health monitoring devices to preprocess physiological signals, reducing the computing burden while providing direct assessments of cardiovascular health. Meanwhile, the memory capability of artificial non-monotonic neurons supports unsupervised learning from accumulated experiential data, enabling adaptive evaluation of health conditions[5]. However,

[1]Department of Electronic Engineering and Materials Science and Technology Research Center, The Chinese University of Hong Kong, Hong Kong SAR, China. [2]Department of Electronics and Nanoengineering, Aalto University, Espoo 02150, Finland. [3]Shenzhen Research Institute, The Chinese University of Hong Kong, Shenzhen 518057, China. ✉e-mail: yqzhou@link.cuhk.edu.hk; jbxu@ee.cuhk.edu.hk

replicating the combined functionality of non-monotonic response curves and memory capabilities in silicon-based devices is both hardware-resource-intensive and energy-consuming[6-8]. For example, well-developed memory devices based on floating gate and interface charge trapping, only demonstrate the monotonic sensing signal coding[9,10]. This is inadequate for handling the non-monotonic nature of biological signals, such as heart rate and blood pressure signals.

The complementary transistors with emerging materials, including two-dimensional (2D) semiconductors[11-13], carbon nanotubes (CNTs)[14,15], and organic materials[16,17], exhibit complex transport characteristics such as negative differential resistance and negative differential transconductance without external ion doping[18-20]. These tunable transfer characteristics enabled the transformation of monotonic input signals into non-monotonic outputs[14,21]. Recent advancements have highlighted the potential of Gaussian synapses based on complementary transistors in advanced computing approaches such as probabilistic inference[22] and analog support vector machine kernel applications[23]. However, these devices rely on multi-gate control, which complicates circuit design, hinders non-volatility integration, and increases energy consumption.

In this study, we manipulated the bell-shaped transfer characteristics of a 2D non-volatile transistor to replicate the non-monotonic response and adaptively learning of non-monotonic neurons. By utilizing tungsten diselenide (WSe$_2$)-palladium diselenide (PdSe$_2$)-molybdenum disulfide (MoS$_2$) junction, we fabricated the anti-ambipolar transistors featuring a benchmarking combination of high peak-to-valley ratio (PVR) of $-10^5$ and subthreshold swings (SS) approaching to 60 mV/dec. Integrating with the floating gate architecture, the individual complementary MoS$_2$ and WSe$_2$ transistors showed high memory speeds ($-10^{-7}$ s), reliable data retention beyond $10^5$ s and robust durability surpassing $10^4$ programming cycles. The anti-ambipolar floating gate transistors (AAFGTs) based on MoS$_2$ and WSe$_2$ transistors can effectively emulate reconfigurable non-monotonic neuromorphic functionalities, including spike-dependent excitatory and inhibitory synaptic behaviors. To show the device's capability of physiological signal analysis, two important indicators of cardiovascular health, SBP and RHR data, were proceed by the non-monotonic neuron as an application example. Through training in the non-monotonic neuron, the accuracy of detection and classification of populations with risk of cardiovascular diseases surpassing 85%, closely corresponding to the recognized medical standards. As a prototypic trial, 2D AAFGT with non-monotonic responses simplify the design while promoting complex behavioral responses.

## Results

### Non-monotonic neurons with non-monotonic response function

Biologically behavioral responses fundamentally operate in two modes: one is a one-to-one mapping from stimulus to behavioral response and the other is a behavioral response triggered only by the intermediate stimulus values. The neuron with an S-shaped response function typically exhibits a monotonic response to presynaptic signals, enabling the monotonic response to stimuli exceeding a threshold[24]. However, the S-shaped response curve poses a challenge for selectively activating signals within an intermediate intensity range, e.g, SBP and RHR. Moderate ranges of SBP and RHR are widely recognized as indicative of healthy cardiovascular function. According to established guidelines, the normal range for RHR is 60-100 beats per minute (BPM), and the normal range for SBP is 90-120 millimeters of mercury (mmHg), as shown in Fig. 1a[25-28]. During the physiological signal processing, the SBP and RHR intensities extracted from monitored signals are monotonically encoded into pulse width and amplitude for integration by subsequent neurons (Fig. 1b). However, using monotonic neurons with S-shaped response function to process these signals results in recognizing only extreme values—those exceeding or falling below the threshold—while failing to identify

signals within the moderate range, as shown in Fig. 1c, d. In contrast, non-monotonic neurons with bell-shaped response functions transform monotonic presynaptic inputs into non-monotonic postsynaptic outputs. This enables selective responses to intermediate stimulus ranges, as illustrated in Fig. 1e. Additionally, these neurons adapt excitatory and inhibitory thresholds through unsupervised learning, avoiding activation range overlap while enabling memory formation and intensity recognition of SBP and RHR signals. (Fig. 1f). AAFGTs as the promising candidate is expected to emulate the behavior of non-monotonic neurons. The bell-shaped transfer curves of AAFGTs demonstrate dynamic tunability, allowing the non-monotonic response curve to be adjusted to recognize specific intermediate-intensity activation ranges. Furthermore, the non-volatility of AAFGTs enables unsupervised learning without the need for external programming, ensuring efficient and adaptive signal processing. These features make AAFGTs ideal candidates for encoding monotonic physiological signals into non-monotonic outputs, providing an alternative information processing approach for the evaluation of potential cardiovascular risk.

### Design of the anti-ambipolar transistor with airgaps configuration

To replicate the non-monotonic response behavior, we designed the WSe$_2$-PdSe$_2$-MoS$_2$ junction anti-ambipolar transistor (AAT) to show bell-shaped channel current ($I_{DS}$)-control gate voltage ($V_{CG}$) curves. The heterojunction transistor is typically configured by connecting MoS$_2$ and WSe$_2$ Schottky barrier transistors. (Fig. 2a) WSe$_2$ and MoS$_2$ transfer characteristics, including the threshold voltage ($V_{th}$), SS, and on-off ratio, contribute to the position and shape of bell-shaped $I_{DS}$-$V_{CG}$ curves, are demonstrated in Fig. 2 and Supplementary Note 1. Compared with traditional P-N junction transistors, such as the WSe$_2$-MoS$_2$ heterojunction, we introduced PdSe$_2$ flake (Supplementary Fig. 2) as a connecting bridge to optimize the symmetry of transfer characteristics of MoS$_2$-WSe$_2$ transistors (Supplementary Fig. 3). WSe$_2$ and MoS$_2$ channels are each in contact with bridged PdSe$_2$, but not contact directly. Optical image of device exhibits in Supplementary Fig. 4. PdSe$_2$ served as both n-type and p-type contact electrodes to avoid the Fermi level pinning effect to ensure the unipolar transfer curves. PdSe$_2$ exhibits high conductivity, low contact resistance, and high electrical stability in the ambient atmosphere (Supplementary Figs. 5 and 6), which contributes to the high stability and endurance of the devices. Kelvin probe force microscope reveal the opposite surface potential difference between PdSe$_2$/WSe$_2$ junction and PdSe$_2$/MoS$_2$ junction as shown in Supplementary Fig. 7a and 7b. The work function (WF) of PdSe$_2$ was also measured to 5.0 eV by using fresh Au film as the reference (Supplementary Fig. 7c). These results indicted the Fermi level of PdSe$_2$ aligns closer with the valence band of WSe$_2$ and the conduction band of MoS$_2$ (Supplementary Fig. 7d). We also extracted the Schottky barrier height (SBH) at the interfaces of PdSe$_2$/MoS$_2$ and PdSe$_2$/WSe$_2$ from low-temperature measurement (detailed calculation is introduced in Supplementary Fig. 8), revealing the n-type and p-type contact barrier heights, with $\Phi_n = 160$ meV and $\Phi_p = 130$ meV, respectively (Fig. 2b). The similar $\Phi_n$ and $\Phi_p$ provide the basis for producing symmetrical n-type and p-type transfer curves. More importantly, in addition to modulating the SBH, we engineered the Schottky barrier width (SBW), which refers to the lateral band bending region in channel formed at the contact area[11]. The corresponding device structure is optimized by incorporating the airgap configuration at the bottom PdSe$_2$ contact. The airgaps are formed at the interfaces between the bottom electrode PdSe$_2$, dielectric h-BN and channels MoS$_2$/WSe$_2$, as shown in Fig. 2c and Supplementary Fig. 9. As reported[29], the triangle-shaped gaps significantly weaken the gate modulation and widen the Schottky barriers' width (SBW), therefore suppress the carrier injection through the bottom PdSe$_2$ electrode into the channel[29]. We investigated the transfer characteristics of PdSe$_2$

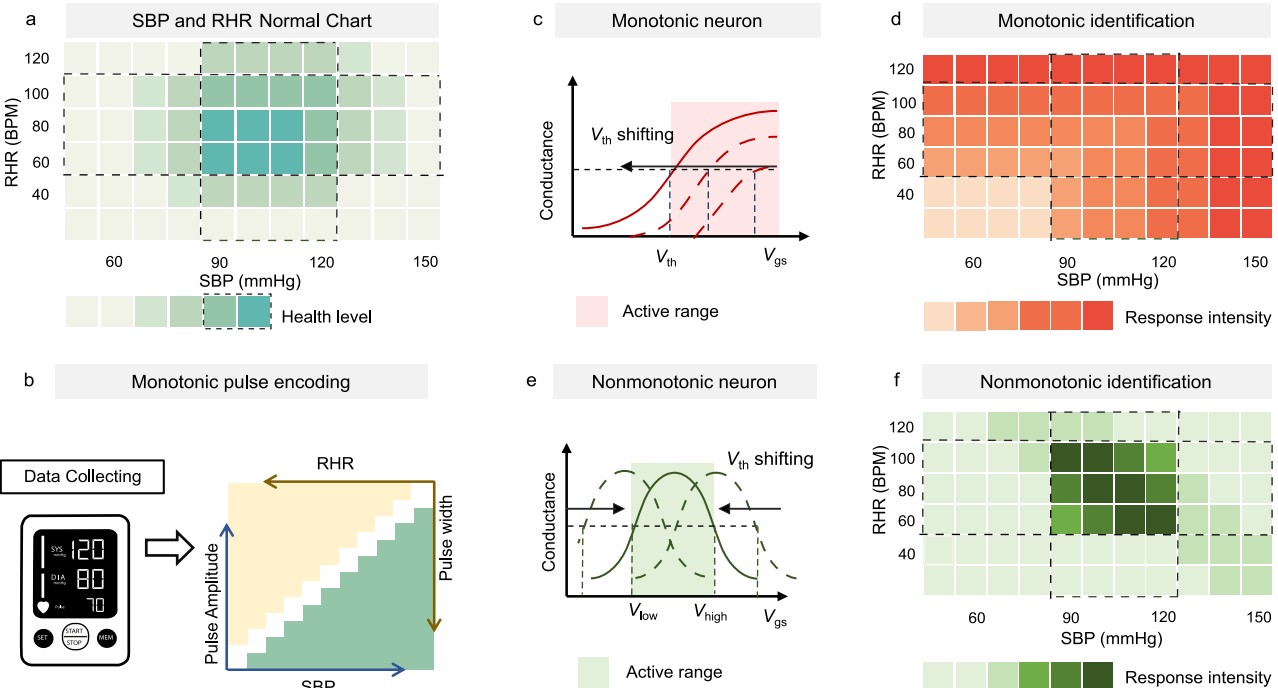

**Fig. 1 | Schematic comparison between monotonic and non-monotonic neurons. a** Distribution of systolic blood pressure (SBP) and resting heart rate (RHR) and their correlation with cardiovascular health levels. **b** Illustration of SBP and RHR data collection and monotonic pulse coding. **c** Schematic of a monotonic excitatory neuron with a *S*-shaped response curve and a shiftable threshold voltage ($V_{th}$). **d** The unidirectional active response range of a monotonic excitatory neuron, lack of selective recognition to intermediate input ranges. **e** Schematic of a non-monotonic neuron with the bell-shaped response curve and a closed response range defined by two parallel shiftable $V_{th}$ values. The dashed regions represent the active response range of the artificial neuron. **f** The closed active response range of a non-monotonic excitatory neuron, showing selectively response to a specific intermediate input range. The shaded regions represent the active response range of the artificial neuron.

contacted transistors with airgaps configurations, in which top PdSe₂ serve as the drain electrodes. The transfer curves in Fig. 2d, e demonstrate successful suppression of ambipolar conversion in both transistors, yielding complementary symmetric unipolar transfer characteristics. As shown in the inset of Fig. 2d, e, electron injection from the PdSe₂ source to WSe₂ and hole injection from the PdSe₂ source to MoS₂ are effectively blocked, therefore the off-state $I_{DS}$ was reduced to considerably low levels of $10^{-5}$ nA. With reversed $V_{DS}$ setting, the transistor showed a lower on-currents due to lower carrier injection efficiency (Supplementary Fig. 10). Compared with transistors without airgap optimization, the $I_{DS}$ on/off ratio increased by three orders of magnitude from $10^3$ to $10^6$ in transistors without airgap optimization (Supplementary Fig. 11). Moreover, MoS₂ and WSe₂ transistors exhibited hysteresis-free transfer curves with near-ideal SS value near 60 mV/dec at room temperature (Fig. 2f and Supplementary Fig. 12), approaching the Boltzmann limit, attributed to the clean and sharp interfaces, as illustrated by HRTEM images in Supplementary Fig. 13. The output characteristics and effective carrier mobility as depicted in Supplementary Fig. 14. Figure 2g illustrates the bell-shaped transfer curves with varying $V_{DS}$, showcasing a low operation voltage of <1.0 V and a narrowed driving voltage range ($\Delta V$) of <1.5 V. The rightward shift of bell-shaped peak is attributed to the stronger electric field distribution over the drain region compared to the source region (Supplementary Fig. 15). The high peak-to-valley ratio (PVR) reaches $1.1 \times 10^5$ at drain voltage ($V_{DS}$) of 1.0 V (Fig. 2h), where PVR represents the ratio of the maximum ($I_{peak}$) and minimum ($I_{valley}$) drain currents. In comparison with the performance of previously reported AATs, our device demonstrates a benchmark PVR with a small $\Delta V$ (Fig. 2i). Additionally, due to its small SS value and low operating voltage, our device achieves a high gain (49 at $V_{DD} = 2$ V) and high operating frequency (~1 kHz) in the inverter mode (Supplementary Fig. 16).

## Nonvolatility of the nonvolatile complementary transistor

The nonvolatile response to input spikes is the prerequisite for achieving specific recognition ability and unsupervised learning capability. We introduced a graphene floating gate in the AAT. Figure 3a shows the transfer curves of the WSe₂ and MoS₂ floating-gate transistors (FGTs) with $V_{GG}$ round sweeping, ranging from ±5 V to ±20 V. The transfer curves exhibit the counterclockwise/clockwise memory window (MW), which increases linearly with the increasing $V_{GG}$ sweeping range and maintains the high on/off ratio over $10^6$, as shown in Fig. 3b. Strong charge transfer typically induces pronounced ambipolarity in 2D FGT, leading to a reduction in the MW and on/off ratio (Supplementary Fig. 17)[30]. Here, however, the airgaps configuration well repressed the ambipolarity, enabling a distinct single counterclockwise or clockwise MW.

Figure 3c shows the $V_{th}$ modulation through the $V_{GG}$ spike amplitude. For the WSe₂ FGT, a $V_{GG} = -15$ V initialized the device in a low resistance state (LRS), and the corresponding $V_{th}$ of the transfer curve was set to 2.1 V (as indicated by the black dotted line in Fig. 3c i). Subsequently, when increasing positive $V_{GG}$, $V_{th}$ gradually shifted leftward to −1.8 V, programming the transistor to a high resistance state (HRS). The reverse $V_{th}$ shift process of the MoS₂ FGT is depicted in Fig. 3c ii), wherein the $V_{th}$ gradually shifted rightward from −3.2 V to 0.3 V, indicating a transition from LRS to HRS. Figure 3d, e exhibit $V_{GG}$ spike width-dependent plasticity of the WSe₂ and MoS₂ FGTs, in which the resistance states were monotonically increased/decreased with the widening of the $V_{GG}$ pulse. Both WSe₂ and MoS₂ FGTs exhibit fast write and erase speeds, owing to the sharp and clean atomic interface and the Schottky barrier assisted Fowler-Nordheim (F-N) tunneling mechanism at drain region[31,32], as detailed in Supplementary Fig. 18. The writing state of WSe₂ FGT could be successfully written at the shortest timescale of 100 ns ($V_{GG} = -10$ V), and it could be erased using a pulse width of 500 ns ($V_{GG} = 10$ V), as extracted in Fig. 3d. For MoS₂

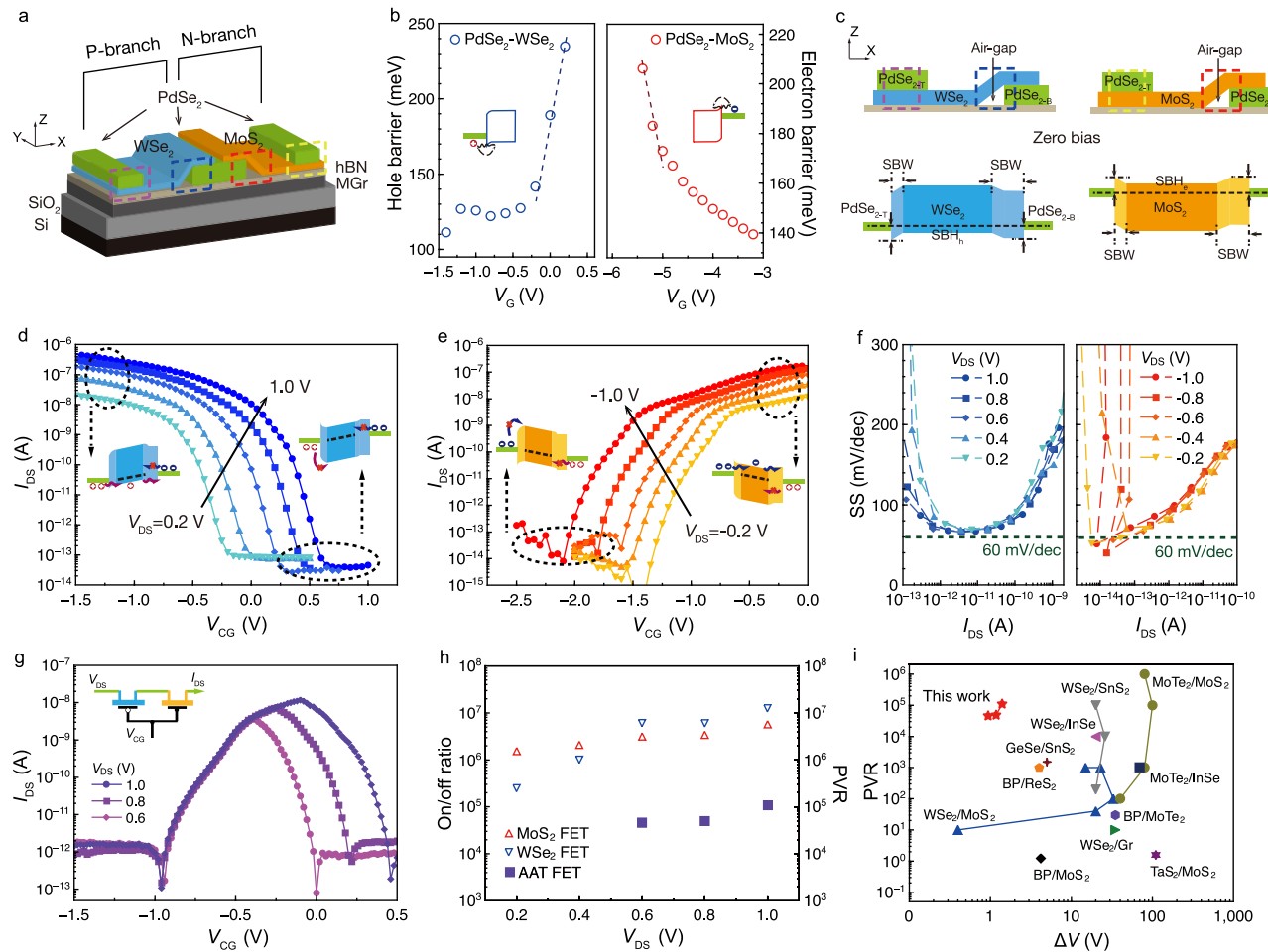

**Fig. 2 | Optimization of junction transistors by using PdSe₂ contacts.**
**a** Schematic of the WSe₂-PdSe₂-MoS₂ junction transistor. The PdSe₂ was alternately arranged between WSe₂ and MoS₂. **b** Flat-band Schottky barrier heights are extracted under gate voltage $V_G$ at flat-band gate voltage $V_{FB}$ conditions. Beyond the $V_{FB}$, the tunnelling current surpasses the thermal emission current, dominating drain current $I_{DS}$. Inset are diagrams of electron and hole barriers. **c** Schematic side views of WSe₂-PdSe₂ and MoS₂-PdSe₂ transistors and their corresponding band diagrams at zero drain bias ($V_{DS}$). The air-gaps only emerged at the bottom-PdSe₂ contact (PdSe₂-B). The Schottky barrier height (SBH) are equal at top-PdSe₂ contact (PdSe₂-T) and PdSe₂-B, while the Schottky barrier width (SBW) at PdSe₂-B is wider.

Transfer curves of PdSe₂ contacted (**d**) WSe₂ and (**e**) MoS₂ transistors with varying $V_{DS}$, sweeping by control gate voltage $V_{CG}$. All transfer curves show unipolar characteristics. **f** Extracted the subthreshold swing (SS) of the complementary transistor. All of them show the near-ideal SS around 60 mV/dec. **g** Bell-shape transfer curves of junction transistors with varying $V_{DS}$, showing a high peak-to-valley ratio (PVR). Inset is the circuit diagram of junction transistors. **h** Comparison of $I_{DS}$ on/off ratio and PVR extracted from transfer curves. **i** The PVR and driving voltage range ($\Delta V$) of the PdSe₂-contacted anti-ambipolar transistor benchmarking the state-of-the-art 2D anti-ambipolar transistors[13,37–53].

FGT (Fig. 3e), the minimum writing pulse width required to alter the resistance state is merely 800 ns ($V_{GG}$ = 10 V), while the minimum erasing pulse is 500 ns ($V_{GG}$ = −10 V). The difference between programming and erasing speeds attributes to the different carrier tunneling barrier heights, as shown in Supplementary Fig. 18. Further, when we use high $V_{GG}$ (Supplementary Fig. 19), the $I_{DS}$ can be fully switched between HRS and LRS (on/off ratio ~ 10⁵) by applying ± 38 V, 400 ns spikes, which closely aligns with the previously reported minimum. Note that the operation speed of our device can be even faster, and the measurement setup was limited by the pulse generator, for which the minimum width of the generated pulse with 38 V amplitude was 400 ns. The ultra-fast write and erase speeds enable the device to exhibit low energy consumption of 19.1fJ and 14.3 fJ for write and erase operations, respectively, which are highly competitive compared to other reported memory devices, as illustrated in Supplementary Table 1.

To ensure reliable nonvolatile memory operation, we examined the retention and endurance performance of multi-bit FGTs, as shown in Fig. 3g, h. We realized 5-state storage in WSe₂ and MoS₂ FGTs. All resistance states show long retention time, and the on/off ratio of

state-0 and state-4 can maintain over 10⁵ after 10⁴ s under continuous measurement at $|V_{DS}|$ = 0.1 V. Meanwhile, the high endurance capability of the WSe₂ and MoS₂ FGTs is demonstrated by repeatedly applying a series of programming and erasing (P/E) voltages, as shown in Fig. 3f. It is evident that both HRS and LRS in the WSe₂ and MoS₂ FGTs showed negligible degradation after 10⁴ P/E cycles. Figure 3i compares the operation speed and endurance parameters of reported 2D FGTs. Complementary devices showed balanced and fast P/E speeds and robust endurances. In particular, the p-type WSe₂ FGT features two orders of magnitude higher speed and longer durability than other p-type memories, significantly improving the compatibility of 2D complementary transistors.

**Spike-dependent plasticity of the non-monotonic neuron**
We further demonstrated the spike-dependent plasticity of non-monotonic neurons based on the complementary FGTs, in which $V_{GG}$ was applied as the presynaptic input and $I_{DS}$ was monitored as the postsynaptic output. The non-monotonic neuron exhibits two distinct $I_{DS}$ peaks during a dual sweep of $V_{GG}$, which is attributed to the synchronous shift of $V_{th}$ in WSe₂ and MoS₂ FGTs. As illustrated in Fig. 4a,

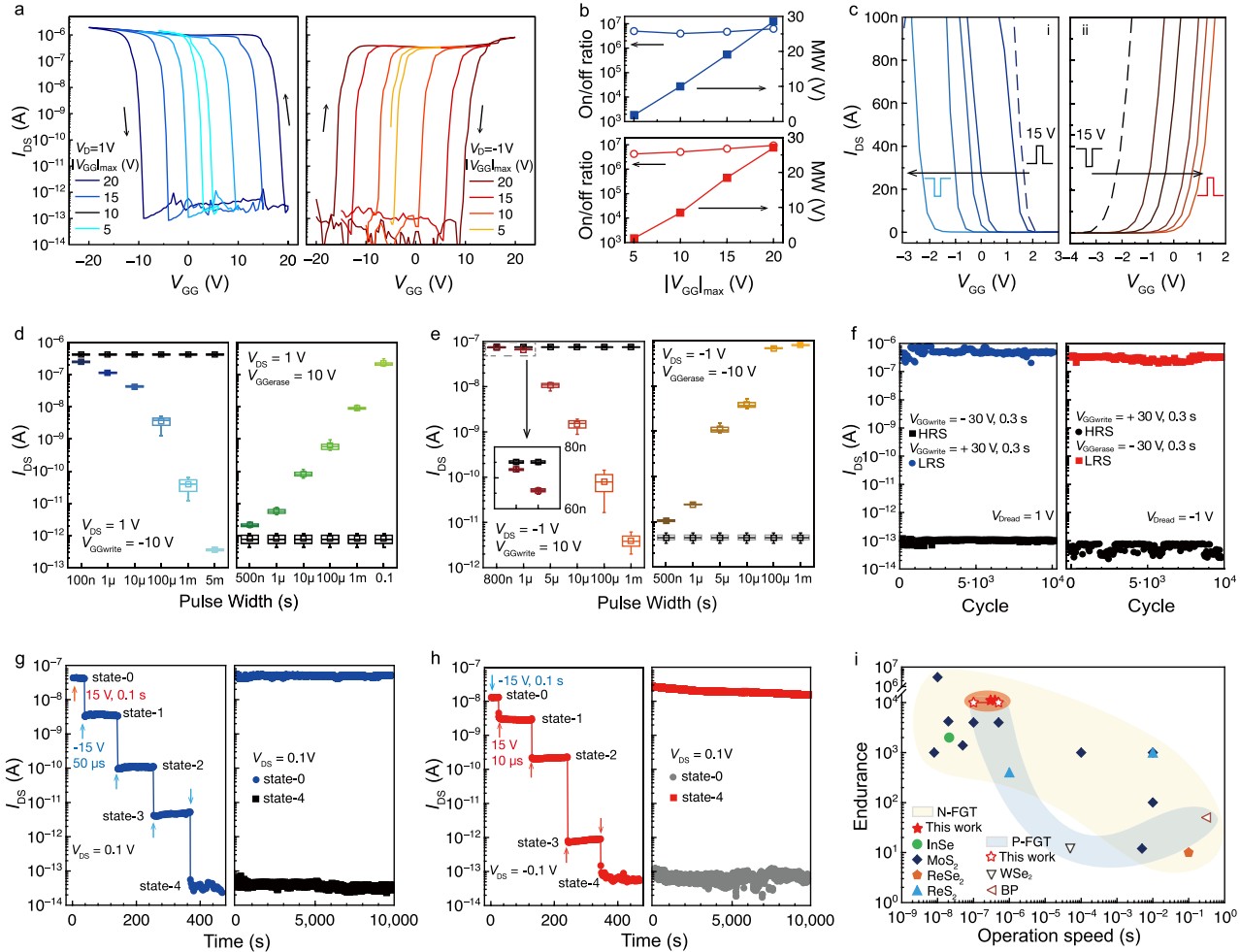

**Fig. 3 | Memory characterization of the n-branch and p-branch in the non-volatile anti-ambipolar transistor. a** Transfer curves measured with global gate voltage $V_{GG}$ round sweeping from negative to positive and back to negative. The maximum $V_{GG}$ ($|V_{GG}|_{max}$) from 5 V to 20 V. **b** Variation of memory window and on/off ratio with varying sweeping range. **c** Threshold voltage $V_{th}$ shifting after different positive $V_{GG}$ spikes were applied. The dash line represents the initial state. **d** Mean value of sampling current of the WSe$_2$ memory after writing/erasing operations under various pulse widths. **e** Mean value of sampling current of MoS$_2$ memory after writing/erasing operations under various pulse widths. **f** 10,000 cycles endurance behavior by applying alternating $V_{GG}$ pulses of 30 V amplitude and 0.3 s width with a reading voltage of $V_{DS} = 1$ V. **g** Multi-level memory states in WSe$_2$ memory are achieved with increasing $V_{GG}$ pulse number. **h** Multi-level memory states in MoS$_2$ memory are achieved with increasing $V_{GG}$ pulse number. Both WSe$_2$ and MoS$_2$ memory show a long retention time beyond $10^4$ s characterized at state-0, state-4. **i** The operation speed and endurance of the PdSe$_2$-contacted complementary FGTs in comparison with the state-of-the-art 2D memory devices[31,32,54–65].

the separation between the forward and backward peaks is enlarged to 14 V under ±10 V $V_{GG}$ sweep. Besides, the $I_{DS}$ peak amplitude can be modulated by $V_{DS}$, as shown in Supplementary Fig. 20. However, increasing $V_{DS}$ also causes a large degradation on the symmetry of the bell-shaped curves. Additional control approach, such as dual-gate modulation, is necessary to facilitate further scaling of $V_{DS}$[12,22,23,33]. Figure 4b demonstrates the $V_{GG}$ programming capability of the non-monotonic neuron. The $I_{DS}$ backward-peak position was first initialized to −5 V. After applying a $V_{GG}$ spike ranging from 2 V to 10 V, the $I_{DS}$ peak position shifted equidistantly rightward to 4 V. This parallel $V_{th}$ movement is mainly attributed to the common graphene floating gate and a uniform h-BN dielectric layer. Despite the fluctuations observed among the $I_{DS}$ peaks (Supplementary Fig. 21) with a standard deviation $\sigma$ of $7.74 \times 10^{-2}$ nA ($\sigma = \sqrt{\sum (x - \mu)^2 / N}$ and, where $x$ is individual peak values, $\mu$ is mean peak value, and $N$ is peak numbers), the PVR is consistently maintained at a value greater than $10^3$.

The tunable peak position allows for a non-monotonic response when the output is acquired at reading voltage of $V_{GG} = 0$ V. As illustrated in the inset of Fig. 4b, the postsynaptic $I_{DS}$ shows a trend of

initial increase followed by a decrease under a series of $V_{GG}$ pulses with varying amplitudes. We evaluated the retention performance by monitoring the output current at $V_{GG} = 0$ V after programming/erasing operations ($V_{GG} = −15$ V/ + 10 V). As shown in Fig. 4c, the peak position shift results in two distinct states, peak (high) and valley (low) output states. Both states demonstrate negligible degradation in both the over $10^4$ s. Cyclic endurance of the AAFGT was characterized by repeatedly measuring the $I_{DS}$-$V_{GG}$ curves with bidirectional $V_{GG}$ sweeping. After 110 cycles of $V_{GG}$ sweeps, the device $I_{DS}$-$V_{GG}$ curves remained closely overlapped, as in Supplementary Fig. 22a and 22b. The switching speeds of AAFGT are provided in Supplementary Fig. 22c and 22d. Compared with the individual WSe$_2$ and MoS$_2$ FGTs, the AAFGT demonstrated a slightly slower speed, achieving successful programming with short pulses width of 500 ns/1 μs ($V_{GG} = ±15$ V). It is worth noting that the data in Fig. 4c and Supplementary Fig. 22 were obtained from a different device with the same structure.

We programmed each 30 same global gate pulses as one set input to the device, Fig. 4d–g demonstrate the non-monotonic encoding capabilities of the neuron for varying input parameters, mainly the width and the amplitude. As shown in Fig. 4d, when different

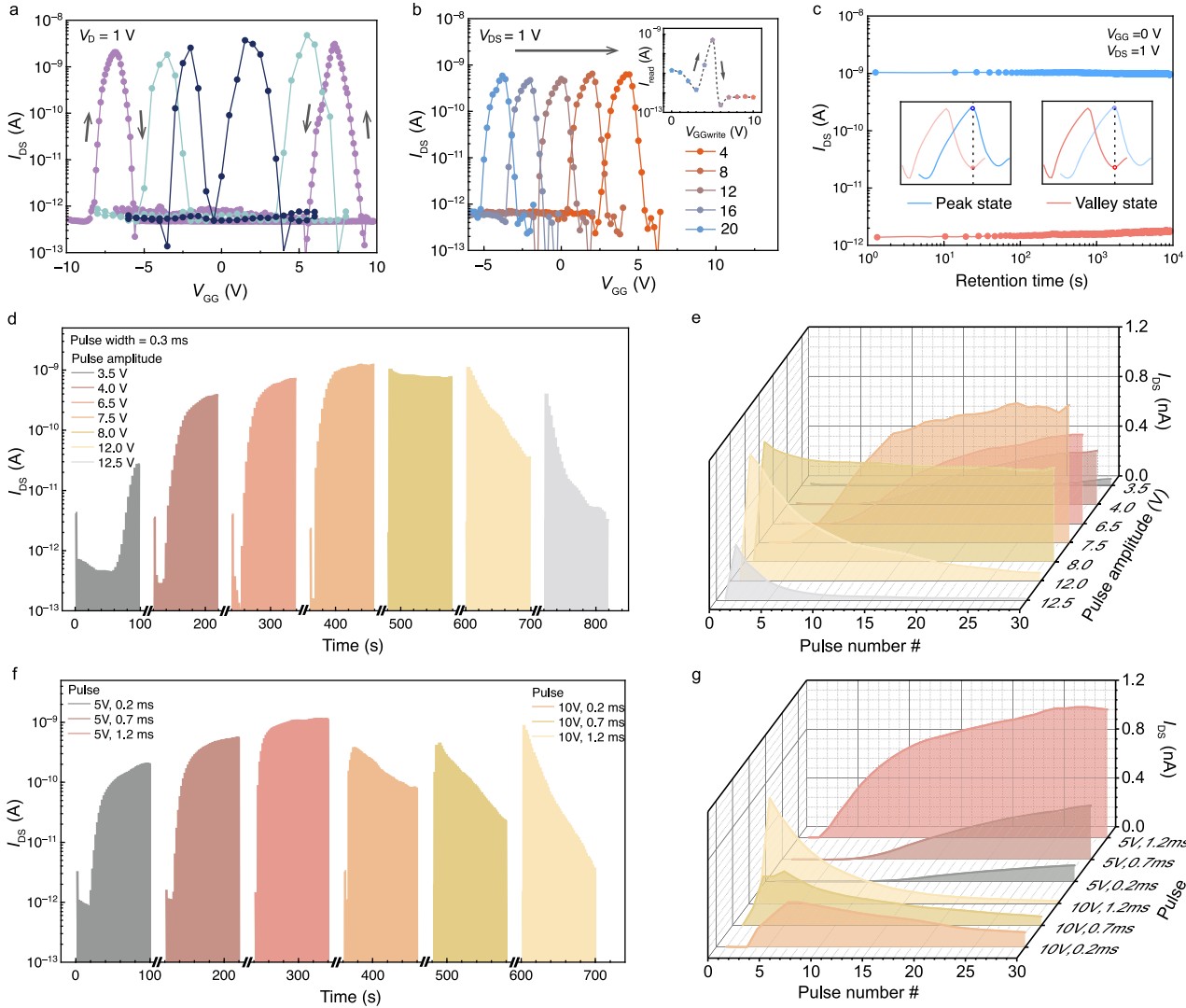

**Fig. 4 | Spike-dependent plasticity of the non-monotonic neuron. a** Peak separation with $V_{GG}$ round sweeping with increasing range $|V_{GG}|_{max}$. **b** Peak shifting after $V_{GG}$ spikes with increasing amplitudes were applied. Inset shows the post-synaptic $I_{DS}$ changing at $V_{DS} = 1$ V and $V_{GG} = 0$ V. **c** Retention of peak and valley current levels over 10,000 s after $V_{GG} = +10$ V/ $-15$ V, respectively. **d** Spike amplitude-dependent plasticity of the non-monotonic neuron. The initial state is reset by the $V_{GG}$ pulse with an amplitude of $-15$ V and a width of 10 ms. **e** Postsynaptic $I_{DS}$ evolution under $V_{GG}$ pulse train with different pulse amplitudes in **d**). **f** Spike amplitude-width plasticity of the non-monotonic neuron. **g** Postsynaptic $I_{DS}$ evolution under $V_{GG}$ pulse train with different pulse settings. All postsynaptic $I_{DS}$ values are measured at $V_{DS} = 1$ V, $V_{GG} = 0$ V. The spike trains including 30 pulses.

amplitudes of pulse sets are applied, the postsynaptic $I_{DS}$ increases for sets of pulse amplitudes below 7 V. At the set with $V_{GG} = 8$ V, the postsynaptic $I_{DS}$ initially surges and then stabilizes at a high level. For those sets with amplitudes greater than 8 V, the $I_{DS}$ initially rises sharply but then starts to decrease, indicative of a non-monotonic response process. The relationship between the pulse number and the output $I_{DS}$ under different pulse amplitudes is summarized in Fig. 4e. Compared with monotonic artificial synapse, which necessitate opposite $V_{GG}$ stimuli to achieve excitation and inhibition, non-monotonic device operates exclusively with identical $V_{GG}$ sign, simplifying the operational setting. Figure 4f illustrates the reconfigurable conductance evolution with varying pulse widths ranging from 0.2 ms to 1.2 ms. Under a low pulse amplitude of 5 V, the sets with wider pulse width only accelerated the activation rate of the synapse. However, for those at larger amplitudes, such as $V_{GG} = 10$ V, both the activation and inhibition rates were accelerated with increasing pulse width, enabling the neuron to exhibit tunable non-monotonic encoding capabilities. The postsynaptic $I_{DS}$ changes with different pulse amplitudes and widths are shown in Fig. 4g. By applying a series of pulse trains with

different amplitude and width, the reconfigurable conductance end-point can be utilized to filter both extremely low (high) values at two ends and moderate values in between. The separation gap between the final memorized states can serve as a quantitative measure for evaluating the distribution range of 'normal' data.

## Automatic encoding of cardiovascular homeostatic features for abnormal detection

Non-monotonic neurons can be utilized for preprocessing and visualizing the SBP and RHR data. By training the neurons with a dataset from normal individuals, we can analyze and visualize SBP and RHR distribution at the device level, enhancing the effectiveness of anomaly analysis and medical diagnostics. Figure 5a shows the workflow for detecting cardiovascular abnormalities using non-monotonic neurons, comprising three modules: **(i)** data preprocessing from physiological measurements, **(ii)** monotonic input generation via a pulse encoder, and **(iii)** individual cardiovascular risk estimation. In module **(i)**, quasi-continuous measurements of physiological parameters, including RHR and SBP, are extracted from the database. These measurements are

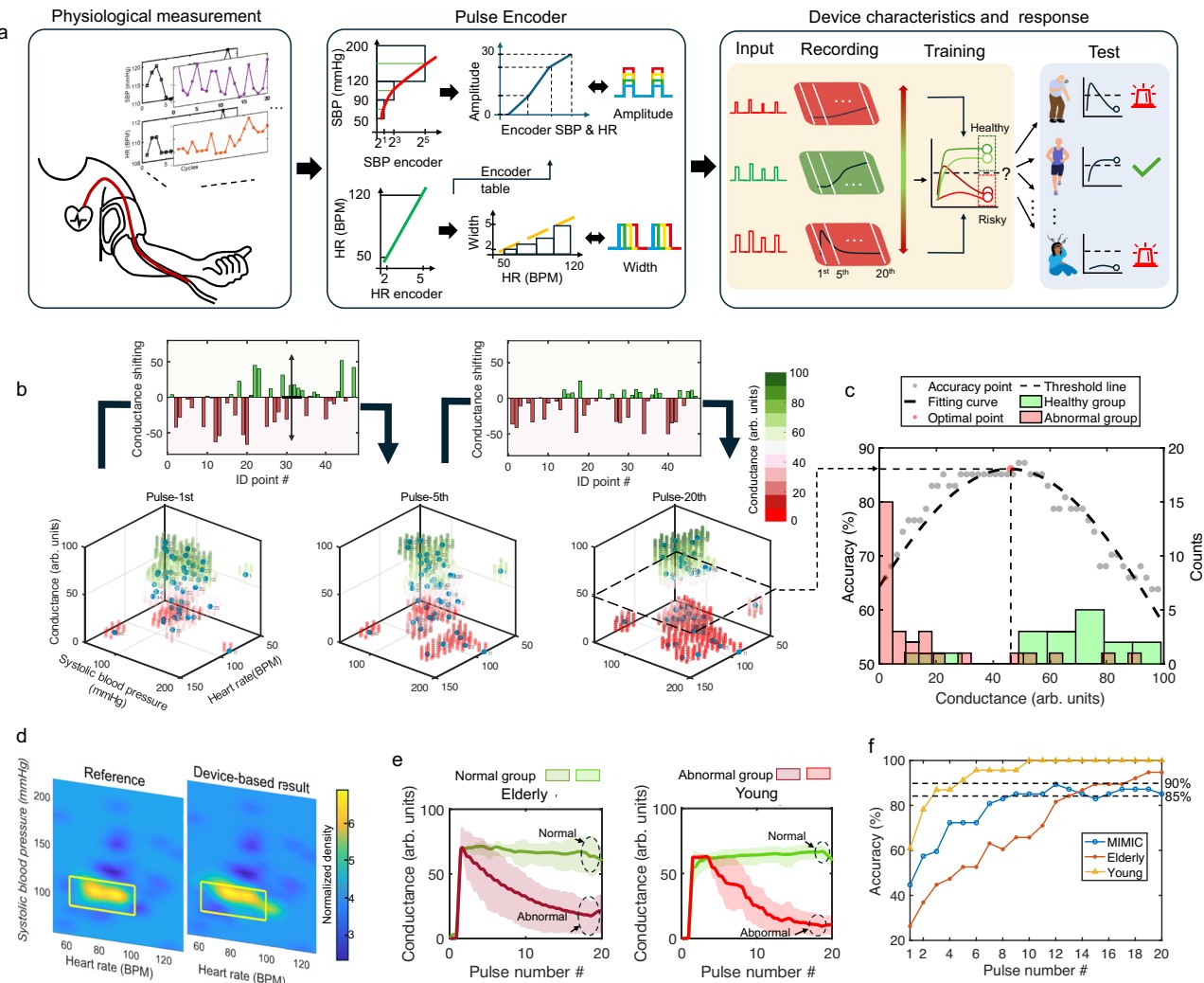

**Fig. 5 | Cardiovascular dysfunction detection using the non-monotonic neuron.** **a** Schematic of the detection process. **i)** Ambulatory rest heart rate (RHR) and systolic blood pressure (SBP) recordings are collected from biosensors. **ii)** Data processing through the encoder. **iii)** Schematic of hyper/hypo-parameter classification enabled by using reconfigurable unimodal response. **b** The device-level conductance responses after the 1st, 5th, and 20th input and the extracted transitional responses for burst pulse input (from 1st to 5th and from 5th to 20th, respectively) in the histogram. **c** Distribution of abnormal data and the calculated classification accuracy. **d** Comparison of the RHR and SBP ranges for healthy individuals extracted via the device with the international standards. **e** Statistical graph of RHR and SBP data distribution among young and elder people. The red and green shaded areas represent the upper and lower quartiles of the device-based abnormal and healthy cardiovascular response curves, respectively. **f** Classification accuracy in three datasets.

then transformed into pulse parameters in module **(ii)** using a homeostasis pulse encoder strategy where the fusion of SBP and RHR is encoded as pulse amplitude and RHR is encoded as pulse width, respectively. The detailed encoding process and classification standard are described in Supplementary Note 2. In module **(iii)**, the encoded pulse patterns are input to the neuron and trained to respond with conductance evolution, enabling the detection of cardiovascular system dysfunctions in individuals via abnormalities (i.e., values above or below the normal range) in RHR and SBP. Unlike conventional physiological diagnosis systems that rely on predefined boundaries for independent abnormal RHR and SBP indices, our non-monotonic-neuron-based assistive diagnosis system integrates and analyzes the boundaries of physiological signal values in-situ through training sets. To be specific, the device-level abnormal threshold detection was carried out by processing real-time clinical recordings from 47 randomly selected subjects from the publicly available MIMIC database (Supplementary Fig. 23) with diverse patient populations[34,35], as shown in Fig. 5b. A total of 47 sets, each consisting of 20 pulses with parameters encoded from the individual's recorded SBP and RHR, were

input into the device, the comprehensive output responses after the 1st, 5th, and the last pulse showing the gradual diverging of normal/abnormal subjects. The distinctions of response current level between the 3 stages are extracted in the upper histogram, where green and red bars represent increasing and decreasing responses, respectively, implying the diversity across subjects' cardiovascular systems. After 20 training cycles, the final response can significantly separate the data points into two distinct extremes. By comparing the responses distribution against established clinical diagnostic standards, we further optimized threshold to achieve the highest classification accuracy beyond 85% (Fig. 5c, f). Figure 5d illustrates the highly consistent response mapping between the device-based and the reference physiological parameters-based classification in RHR and SBP, as indicated in the dashed box. The results demonstrate the non-monotonic neuron is capable of reliably unsupervised learning and classifying individuals based on their physiological indicators.

After training using the clinical external dataset, we evaluated the system's reliability and stability using internal-built PWH datasets from our previous research[36], which included different age groups (PWH-

Elderly and PWH-Young, see Supplementary Table 2). This approach was applied to cardiovascular system detection based on the measured RHR and SBP in both young and elderly populations. 61 records (38 the elderly and 23 the young) from the geriatric care hospital and lab, respectively were used as the input dataset to the non-monotonic neuron (Supplementary Fig. 24). A significant decreasing trend of the abnormal cardiovascular systems (the solid red color line with upper and lower quartiles of the device-based response in the red shaded region) was detected when compared with the subject with normal cardiovascular systems (the solid green color line with corresponding device-based response curve in the green shaded region) in both young and elderly population in Fig. 5e. The detection accuracy in both age groups were higher than 90% (Fig. 5f), demonstrating the capability of the device for large-scale, all-age group analysis. The ability to maintain a consistent accuracy across diverse age groups indicates the robustness and reliability of the device in clinical settings.

## Discussion

Traditional memory devices with monotonic response functions are inherently limited in their ability to replicate the distinct non-monotonic response characteristics of neurons. Herein, we manipulated the non-volatile bell-shaped transfer characteristics of 2D heterojunction transistors to replicate specific responses to the intermediate range and unsupervised learning of non-monotonic neurons. We arranged $PdSe_2$ alternately in the $MoS_2$ and $WSe_2$ junction to optimize the bell-shaped transfer curve of the transistor. Specifically, $PdSe_2$ as the top contact can avoid the Fermi level pinning at the interface between metal and 2D semiconductor, forming p-type and n-type Schottky barriers with $WSe_2$ and $MoS_2$, respectively, and achieving precise control of their transfer curves. Meanwhile, the introduction of airgaps by $PdSe_2$ as the bottom contact can also suppress the off-state $I_{ds}$ currents. These optimization strategies enable the junction transistor to show a benchmark combination of high PVR of ~$10^5$ and near-ideal SS of 60 mV/dec. We fabricated the individual $WSe_2$ and $MoS_2$ FGTs showing robust multi-bit storage with high operation speed at ~$10^{-7}$ s level and reliable endurance of > 10000 cycles. Based on this strategy, we further designed a reconfigurable non-monotonic neuron showing a reconfigurable $V_{GG}$ pulse-dependent nonvolatile response to emulate the synaptic behaviors. By encoding the amplitude and width of presynaptic $V_{GG}$ pulse with the SBP and RHR data from the MIMIC database, we achieved classification of healthy individuals and prediction of cardiovascular conditions with a detection accuracy surpassing 85% at the device level. The 2D non-volatile anti-ambipolar transistor is expected to significantly simplify the system complexity involved in implementing behavioral responses.

## Methods
### Device fabrication
All the 2D materials were prepared by the mechanical exfoliation method and the heterostructure-based AAT was stacked via the multi-layer-by-layer dry transfers. Exfoliated few-layer graphene and h-BN were stacked on the poly(propylene) carbonate (PPC) film supported by the polydimethylsiloxane (PDMS) stamp. The graphene/h-BN junction was firstly dry transferred onto the highly doped p-type silicon substrate. Similarly, exfoliated $WSe_2$, $PdSe_2$, and $MoS_2$ were stacked on the surface of a PPC film and dry transferred on the surface of a graphene/h-BN junction in sequence. The PPC residues were removed with acetone. Afterwards, the electrodes of Au/Cr (50/10 nm) were defined by electron-beam lithography and deposited by electron beam evaporation on the 2D heterostructures. If necessary, top graphene and h-BN were also assembled and dry transferred to the top surface and washed off the residue with acetone.

### Material characterization
AFM (Bruker, Dimension Icon) in the tapping mode TUNA mode was employed to measure the thickness of the device, while the contact potentials of the different areas were measured via the Kelvin probe force microscopy. Micro-Raman investigation was performed using HORIBA LabRAM HR Evolution system with 532 nm laser excitation (the laser spot was ~1 μm in diameter). The cross-sectional characteristics of heterostructures were prepared by a focused ion beam (Thermoscientific Scios 2) and measured via aberration-corrected high resolution transmission electron microscope (FEI Tecnai F200 systems). The element distribution was analyzed by energy-dispersive spectrometer.

### Electrical characterization
The devices were tested in a Cascade probe station under high vacuum conditions. The electrical measurement was performed through the Keithley 4200 semiconductor characterization system. The high-speed memory characterization of the device was executed with the global gate terminal connected to the PMU unit equipped in 4200a-SCS as input signal, while the drain and source terminals were connected to the 4200-SCS SMU. The grounds of the PMU and SMU units were connected.

### Cardiovascular dysfunction detection
Both SBP and RHR were extracted from the peaks and the beat-to-beat peak intervals of the arterial pulse waveform, respectively. The clinical standard ranges for SBP and RHR are referenced from the guidelines established by the American Heart Association and the European Society of Hypertension[25–27]. According to these guidelines, the normal range for resting heart rate is 60-100 beats per minute (BPM), and the normal range for blood pressure is 90-120 millimeters of mercury (mmHg). These physiological parameters exhibit a typical non-monotonic property due to the regulation of the cardiovascular homeostatic system. To enable device-level cardiovascular dysfunction detection, a pulse encoder strategy was proposed to align SBP and RHR with the device input characteristics. The amplitude of the device input (mixed coding by SBP and RHR in Fig. 5a) mimics the cardiac contraction intensity (i.e., feedforward modulation) of the cardiovascular system, which involves the strong connection between SBP and RHR due to the cardiovascular coupling effect[28]. Conversely, the width of the device input (shown in Fig. 5a) mimics the cardiovascular venous returning (i.e., feedback modulation) that is encoded linearly by the RHR within a range of[25,26] for the width of the pulse input[28]. The device responds to the encoded pulse inputs and may adapt to a stable trend for further detection of cardiovascular dysfunction. The optimal threshold point (OTP) was identified through an examination of the response of device conductance, aligning it with the minimizing the binary cross-entropy loss (i.e., highest accuracy) of the ground truth classification of cardiovascular systems, by utilizing an external public MIMIC dataset[34,35], i.e.,

$$OTP = \underset{c}{\arg\min} \, L\left(C_{ref}, C_{dev}\right) = -\left(C_{ref} * \log\left(C_{dev}\right) + (1 - C_{dev}) * \log\left(1 - C_{ref}\right)\right)$$

(1)

where $C_{ref}$ is the reference label of the cardiovascular system determined by medical guideline[25–27]; $C_{dev}$ is the predicted label of the cardiovascular system by applying the determined conductance (c) threshold to the device responses. Finally, the internal-built databases[36] were used to further validate the clinical performance of the device. The experiment was approved by the Joint Chinese University of Hong Kong–New Territories East Cluster Clinical Research Ethics Committee and received consent from the participants (Joint CUHK- NTEC CREC Ref. No.: 2022.335). Note that when participants are older than 65 or younger than 35, subjects will be categorized as the elderly and young groups, respectively. Detailed datasets are summarized in Supplementary Table 2.

## Data availability
Relevant data supporting the key finding of this study are available within the article and the Supplementary Information file. All raw data

generated during the current study are available from the corresponding authors upon request. Source data are provided with this paper.

## Code availability

The code that supports the finding of this study is available from the corresponding author upon request.

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

## Acknowledgements
The work is in part supported by Basic and Applied Basic Research Foundation of Guangdong Province (No. 2023B1515120049), Research Grants Council of Hong Kong (Grant No. 14212424), and CUHK Group Research Scheme, RGC Postdoctoral Fellowship, CUHK Postdoctoral Fellowship.

## Author contributions
Y.P., Y.Z. and J.X. conceived the idea. Y.P. and Y.Z. designed and fabricated the devices. S.Q., Y.P. and Y.Z. designed the encoding method and provided the biological model. L.T. and N.Z. contributed to the discussions. Y.P., S.Q. and Y.Z. analyzed the data and wrote the manuscript. All authors discussed the data analysis and commented on the manuscript.

## Competing interests
The authors declare no competing interests.
