## [Transparent Peer Review file · Nature Communications]

Artificial non-monotonic neurons based on nonvolatile anti-ambipolar transistor

Corresponding Author: Professor Jianbin Xu

Version 0:

Reviewer comments:

Reviewer #1

(Remarks to the Author)

This work reported by Pang Y. et al. demonstrates non-monotonic neurons by utilizing 2D heterostructures with nonvolatile bell-shaped transport features. The replication of specific responses to the intermediate range and unsupervised learning has been achieved. The introduction of PdSe₂ and airgap facilitates the optimization of device performance. The device exhibits pulse-dependent reconfigurable nonvolatile response with fast operation speed, which enables the non-monotonic synaptic behaviour that is beneficial to the analysis of physiological data. However, some claims lack of experimental supports, and there are several concerns need to be addressed.

1. In Figure 1, the schematics of amplitude vs. time do not show explicit correspondence to the upstream signals. The authors are suggested to clarify this point. Moreover, why the learning process is manifested as the shift of V_{th} ? How the specific response range is defined, is it related to the upstream signals?
2. In page 5, how does the KPFM demonstrate relative alignment between the PdSe₂ work function and valence/conduction band of the channel materials? Also, the airgap does not seem to be an intentional optimization, but rather an occasional and uncontrollable factor during the transfer process. The authors should have more detailed discussions on the optimization effect of this gap and how to control its features based on their claim of structural optimization.
3. In Figure 2, the authors are suggested to provide a clearer illustration of the device structure. Do the MoS₂ and WSe₂ touch by edge? Moreover, it would be better to have explicit comparisons between the transport properties with and without the insertion of PdSe₂, as well as the airgap.
4. In Supplementary Figure 10, is the airgap presented in the schematics? How does it affect the tunneling process? In Figure 3d, why the writing and erasing speed in WSe₂ differ a bit large? In Figure 3e, why the MoS₂ channel exhibits a slower operating speed comparing to WSe₂? Can the authors provide calculations on the programming energy consumption and compare with other devices with similar function?
5. In Supplementary Figure 14, the authors should explicitly describe the standard for determining the abnormal and normal cardiovascular system. For example, why id 22 is abnormal but id 21 is normal despite they look the same? And why id 32 is normal but not passing through the threshold?
6. There is an incomplete sentence in the abstract line 21, "Finally, to highlight the device's application". In Figure 2e, the y-axis unit is supposed to be A. Figure 2i should be 2h. The pulse axis in Figure 4g seems mistakenly reversed comparing to Figure 4f.

Reviewer #2

(Remarks to the Author)

In this study, an anti-ambipolar transistor, composed of a P-type WSe₂ and N-type MoS₂, incorporates a PdSe₂ bridge between the two semiconductors to create an air gap structure. This setup suppresses ambipolarity and prevents Fermi level pinning using PdSe₂ electrodes, ensuring unipolar driveability. Additionally, a graphene floating gate is inserted at the front

to enable memory operations. By adjusting V_d , a transfer curve was produced, whose size can be modulated and horizontally shifted through programming. Using this device, a new form of spike-dependent plasticity was proposed for non-monotonic neurons. When SBP and RHR were processed as the amplitude and width of input pulses, respectively, and 20 learning cycles were conducted with these neurons, the system could classify normal and abnormal cases with an accuracy of 85%.

Although this study is quite interesting in that authors implements non-monotonic neurons by producing high-performance anti-ambipolar transistors using inorganic semiconductors, I feel further consideration is necessary, Especially some concerns must be addressed before its publication, which are noted below.

Minor comments;

- It is necessary to check for a typo on page 3, line 3, where "random blood pressure (SBP)" might have been written incorrectly.
- In Figure 3-a's transfer curve, it would be better to choose a color combination with greater contrast for the four types of VGG max in order to more clearly demonstrate the characteristic of increased hysteresis during double sweep.
- In the caption of Supplementary Figure 12, there are two instances of "b)." A typo correction is needed.
- For Supplementary Table 1, clarification is needed on whether the PWH-Elderly and PWH-Young datasets are separate from the MIMIC I dataset. If they are separate, an explanation is required on how these datasets were extracted (if they are datasets from previous studies, references seem necessary).

Major Comments

- In the introduction, the explanation of artificial neurons is quite lacking, and the explanations of non-monotonic and monotonic neurons are also insufficient. Additionally, moving parts of the SBP and RHR content from the Supplementary Note to the introduction or the clinical data categorization section would aid reader comprehension. The classification of data using SBP and RHR for normal/abnormal conditions seems to appear too abruptly.
- On page 7, line 7, there is no explanation of the VGG double sweep, making it difficult to understand. Without this explanation, Figure 3a could lead to misunderstanding, making it seem like the V_{th} changes without reason.
- On page 7, line 11, the phrase "air gaps near the bottom PdSe2 bridge well repressed the ambipolarity, resulting in a single counterclockwise or clockwise memory window" is used. If there is data showing the transfer characteristics without the air gap in this device structure, it would make it easier for readers to understand how ambipolar conversion and off-current suppression are achieved.
- In Figure 3, while there is an overall explanation of the memory characteristics of P-type and N-type single devices, it would be reasonable to use single devices for multistate (MW) measurements. However, for retention, endurance, and speed, there could be significant performance differences between AAT memory devices and single semiconductor memory devices. Therefore, memory characteristics should be presented for AAT devices, not single semiconductor devices.
- On page 9, line 21, it is mentioned that the peak of the anti-ambipolar transfer curve can be adjusted by controlling VDS, as shown in Supplementary Figure 12, thereby increasing the tunability of the neuron. However, if we refer to Supplementary Figure 12, the peak differences are not significant. It seems necessary to remeasure with a larger difference in VDS. Additionally, when adjusting VDS, not only the peak but also the threshold voltage in the P-type current region changes. Therefore, it should be clearly stated that perfect scaling is not achievable by adjusting VDS alone.
- On page 9, line 25, it is stated that the transfer curve's peak height can be maintained while shifting the peak position through programming. For the peak height to remain constant, the threshold voltages of the P-type and N-type semiconductors must move by the same amount through programming, which is very difficult to achieve. Additionally, in Figure 4C, it appears that the peak height remains constant because the graph is plotted on a log scale, but it could show significant differences if plotted on a linear scale. Since maintaining a consistent peak height is critical when using this curve for neurons, Figure 4C should be presented on a linear scale, and data showing consistent peak height should be added.
- In Figure 5b, it would help readers understand why the MIMIC dataset was selected, what kind of data it is, and what processing was done on the raw data. This should be specified more clearly in the Supplementary Note. Additionally, bibliography data for references 6 and 7 are missing from Supplementary Table 1 (these seem to be MIMIC-I references).
- In Supplementary Figure 4, when extracting the Schottky barrier, it is indicated that the same polarity of drain voltage was used for both the electron and hole sides, but different polarities should be used. This needs to be rechecked. Additionally, the equation for the current density used in the thermionic emission regime for barrier height extraction should be added to the explanation of Supplementary Figure 4.
- In the equation for setting pulse width in Supplementary Note 2, if the resting heart rate data exceeds a certain range, it seems to go beyond the suggested manipulation width. Additional explanation is needed for the basis of this equation. The

same applies to the construction of the SBP encoder equation below. Providing more examples and explaining why these settings were made would help readers understand better.

Reviewer #3

(Remarks to the Author)

The authors reported a 2D anti-ambipolar transistor with a floating gate structure, which functions as a non-monotonic neuron, demonstrating unsupervised learning and intermediate recognition capabilities. Experimentally, the authors employed this non-volatile 2D anti-ambipolar transistor to process physiological data, including blood pressure and resting heart rate, achieving device-level predictions of cardiovascular risk. The concept of non-monotonic response is intriguing, and the device exhibited a benchmark high PVR of ~105 and symmetric n and p type non-volatile properties. However, some minor concerns need to be addressed before publication

The detailed question is below:

- 1, In the introduction of the non-monotonic neuron, the authors should provide a clear explanation of Figure 1, including the significance of the colored shaded areas, and explain the advantages and necessity of using non-monotonic neurons to process SBP and RHR data. This will make the concept more intuitive.
- 2, In Figure 1d, the V_{th} of both WSe₂ and MoS₂ transistors could be shifted by VDS. However, in Figure 1e, the V_{th} at left side remains unchanged. It is essential to explain the difference in the V_{th} shift in anti-ambipolar transistor with VDS.
- 3, On page 5 line 15, the text mentioned that the air-gap occurs at the bottom PdSe₂ electrode contributes to optimizing the transfer characteristics. The schematic should highlight the air-gap and provide more detailed discussion on how this configuration influence the device performance. Including transfer curves with reversed VDS is crucial. Additionally, the authors should clearly explain the methodology used to calculate the Schottky barrier.
- 4, The anti-ambipolar transistors in previous report require the coordination of drain and dual-gate voltages to achieve parallel shifts of the IDS peak position and maintained peak height [Nat Commun 11, 1565 (2020)]. The authors should discuss why, in floating-gate-based transistors, parallel shifts of the peak can be achieved using a single gate.
- 5, In Figure 5a and Supporting Note 2, different encoders are employed for SBP and RHR data, respectively. it is necessary to give a more detailed explanation to the choose of encoding models.
- 6, For the application, the authors should clearly introduce the criteria used to distinguish between healthy individuals and those at risk, along with the corresponding references.

Version 1:

Reviewer comments:

Reviewer #1

(Remarks to the Author)

The authors have addressed all my questions and comments.
I would like to recommend the publication in Nature communications.

Reviewer #2

(Remarks to the Author)

The authors have done an excellent job of reflecting the reviewers' comments, conducting additional revision experiments with their full devotion, and revising the text accordingly. There are still a few points that could be improved. Therefore, we would like to provide the following minor suggestions:

1. In the revision, we requested experiments on memory characteristics in anti-ambipolar devices rather than p- and n-type unit devices. While the retention characteristics of the original unit devices were shown for 10,000 seconds, the revised file only shows 2,500 seconds for the anti-ambipolar device. It would be better if a plot extrapolated to a longer time frame could be provided.

2. The section on Schottky barrier width (SBW) has been revised, but the concept of SBW has not been clearly defined. It is unclear what specific part of the semiconductor at the contact is being referred to as SBW. Additionally, the references cited in this section do not explicitly define Schottky barrier width. Providing a more detailed mathematical or conceptual explanation of SBW in the manuscript would help readers understand this concept more clearly.

Reviewer #3

(Remarks to the Author)

the authors had addressed all the concern.

**Response to reviewers' comments**

**Response to reviewer 1#**

**General comments:**

*“This work reported by Pang Y. et al. demonstrates non-monotonic neurons by utilizing 2D*
*heterostructures with nonvolatile bell-shaped transport features. The replication of specific*
*responses to the intermediate range and unsupervised learning has been achieved. The*
*introduction of PdSe₂ and airgap facilitates the optimization of device performance. The device*
*exhibits pulse-dependent reconfigurable nonvolatile response with fast operation speed, which*
*enables the non-monotonic synaptic behaviour that is beneficial to the analysis of physiological*
*data. However, some claims lack of experimental supports, and there are several concerns need*
*to be addressed.”*

**Response:**

We sincerely thank the reviewer for their valuable comments and suggestions. In the revised
manuscript, we have added experimental demonstrations and detailed discussions to verify the
working mechanisms of the anti-ambipolar transistor utilizing airgap configuration and its
application in physiological information processing. We have also carefully reviewed the
manuscript and figures, corrected typographical errors. **We incorporated point-by-point**
**responses to the reviewer’s concerns. The revisions have been marked in red in the**
**manuscript.**

The main revisions and responses are summarized as follows:

- ➤ We have revised **Introduction** section and **Figure 1** to explicitly illustrate the
relationship between pulse encoding and upstream physiological signals for enhanced
clarity.
- ➤ We have added a detailed explanation of the surface potential data measured by Kelvin
Probe Force Microscopy (KPFM) to better elucidate the device’s operating principles.
Additionally, we have included the detailed description of the air-gap structure and a
comprehensive discussion on the role of the air-gap strategy.

- ➤ We fabricated and compared the performance of WSe₂-MoS₂ junction transistors with
and without the PdSe₂ bridge to elucidate the critical role of the PdSe₂ bridge and
airgaps configuration in improving device performance.
- ➤ We have added a detailed discussion on the tunneling path. We have also explained the
origin of the inconsistent programming and erasing speeds observed in the WSe₂ and
MoS₂ non-volatile transistors based on the Schottky barrier-assisted tunneling
mechanism.
- ➤ We have provided a detailed calculation of the programming energy consumption of
our memory devices and compared it with previously reported memory devices,
demonstrating the energy efficiency of our devices.
- ➤ We have thoroughly reviewed the manuscript for typographical errors and made the
necessary corrections.

We hope that the revised manuscript addresses all the reviewer’s concerns and enhances the
clarity and quality of our work. Thank you once again for your valuable feedback.

**Comment 1:**

*“In Figure 1, the schematics of amplitude vs. time do not show explicit correspondence to the*
*upstream signals. The authors are suggested to clarify this point. Moreover, why the learning*
*process is manifested as the shift of V_{th} ? How the specific response range is defined, is it*
*related to the upstream signals?”*

**Response:**

We appreciate the reviewer’s constructive comment. Following the suggestion, we have revised
**Figure 1** to explicitly illustrate the relationship between pulse encoding and upstream signals.
Specifically, the systolic blood pressure (SPB) and resting heart rate (RHR) value, monitored
using a commercial healthcare monitor, are monotonically encoded into gate voltage V_{GS} pulses
with varying amplitudes and durations, as depicted in the revised **Figure 1b**. This encoding
process establishes a direct correspondence between the upstream bio-signals and the input
pulses.

We also clarified how the learning process manifested through the shift of threshold voltage
 (V_{th}). In our device, the two V_{th} on left and right sides (V_{th-l} and V_{th-r}) collectively determined
 the peak position (V_{peak}), as shown in the **Figure R1 a**. While the bell-shaped ambipolar transfer
 (AAT) curve maintains a constant width, its position shifts due to changes in V_{th} . This shift leads
 to variations in the response current (I_{read}) at the fixed reading voltage (V_{read}), as shown in **Figure**
 **R1 a-c**. When the two V_{th} values bracket V_{read} ($V_{GG} = 0$ V in **Figure R1 a**), the I_{read} reaches its
 maximum, representing the high response. As the V_{th} values shift either left or right away from
 V_{read} , I_{read} decreases rapidly, leading to a reduced response. (**Figure R1 b and c**) Hence, the
 positions of V_{th} directly govern the response intensity of our device, which should be optimized
 during the learning process.

Furthermore, the specific response range is determined by the upstream training signals and
 threshold line optimization in the learning process. In our study, 47 sets of blood pressure and
 heart rate data were used as the training dataset and encoded into pulse trains with varying
 amplitudes and widths, which were subsequently applied to our device. These training pulses
 induced in V_{th} and I_{DS} peak shifting. Following pulses training, we optimized the I_{DS} response
 thresholds line by comparing with the training database (**Figure 5b and 5c**) and achieved
 classifying individuals based on their physiological indicators.

**Figure R1** Peak position (V_{peak}) shifting controlled by threshold voltages (V_{th}) for the varying
 current response states. **(a)** $V_{peak} \approx$ reading voltage (V_{read}). **(b)** $V_{peak} \gg V_{read}$, **(c)** $V_{peak} \ll V_{read}$.

**The corresponding discussions in the revised manuscript:**

“During the physiological signal processing, the SBP and RHR intensities extracted from monitored
 signals are monotonically encoded into pulse width and amplitude for integration by subsequent
 neurons (**Figure 1b**).” (Page 3, line 27)

“Unlike conventional physiological diagnosis systems that rely on predefined boundaries for
 independent abnormal RHR and SBP indices, our non-monotonic-neuron-based assistive
 diagnosis system integrates and analyzes the boundaries of physiological signal values in-situ
 through training sets. By comparing the responses distribution established clinical diagnostic
 standards, we further optimized threshold to achieve the highest classification accuracy beyond 85%
 (Figure 5c and 5f).” (Page 14, line 12)

**Comment 2:**

“In page 5, how does the KPFM demonstrate relative alignment between the PdSe₂ work
 function and valence/conduction band of the channel materials? Also, the airgap does not seem
 to be an intentional optimization, but rather an occasional and uncontrollable factor during
 the transfer process. The authors should have more detailed discussions on the optimization
 effect of this gap and how to control its features based on their claim of structural optimization.”

**Response:**

Thanks for the reviewer’s valuable comments and suggestions. We have added the detailed
 explanations of surface potential data measured by KPFM. In the KPFM measurement, the
 work function (surface potential) Φ_{2D} of 2D materials is measured by the **Equation 1**:

$$18 \quad \Phi_{2D} = eV_{CPD} - \Phi_{tip} \quad \text{Eq.1}$$

Where e is the electron charge and V_{CPD} is the contact potential difference between the Φ_{2D} and
 the work function of conducting probe Φ_{tip} . Since Φ_{tip} remains constant during measurement
 during measurement, the work function difference $\Delta\Phi_{2D}$ between adjacent material is calculated
 by the **Equation 2**:

$$23 \quad \Delta\Phi_{2D} = e\Delta V_{CPD} \quad \text{Eq.2}$$

According to our KPFM result, the PdSe₂/WSe₂ junction showed a positive $\Delta V_{CPD} = 45$ meV,
 indicating that Φ_{PdSe_2} is higher than Φ_{WSe_2} , while the PdSe₂/MoS₂ junction showed a negative Δ
 $V_{CPD} = 52$ meV, indicating the Φ_{PdSe_2} is lower than Φ_{MoS_2} , as shown in **Figure R2 a**. Besides, we
 acquired absolute $\Phi_{PdSe_2} = 5.0$ eV via utilizing fresh Au film ($\Phi_{Au} = 5.1$ eV) as the reference as
 shown in **Figure R2 b**. By comparing the electron affinity χ_S and band gap E_g for multi-layered
 MoS₂ and WSe₂ ($\chi_{MoS_2} = 4.4$ eV, $E_{g, MoS_2} = 1.3$ eV; $\chi_{WSe_2} = 3.7$ eV, $E_{g, WSe_2} = 1.7$ eV),¹⁻³ a band

alignment diagram was constructed in **Figure R2 c**, demonstrating that the PdSe₂ work function
aligns near the valence band of WSe₂ and the conduction band of MoS₂.

The airgap is common in stacked 2D junctions formed via the dry transfer method, with
its size adjustable by modifying the 2D material thickness. We expanded on the airgap strategy
in the PdSe₂-contacted transistor. According to the general Schottky barrier theory of the 2D
transistor,^{4,5} the gate modulation will induce the strong electrostatic doping and contact barrier
thinning, enabling the carrier to be injected into channel from both the source and drain
electrode, which is the origination of the ambipolar transfer characteristics in 2D transistor, as
shown in **Figure R2 d-f**. In this work, we used the air-gap strategy to suppress ambipolar
transfer characteristics by widening the contact barrier and suppress the carrier injection from
source electrode.⁶⁻⁸ Specifically, the triangular air gap is formed by bottom PdSe₂ electrodes,
MoS₂ and WSe₂ channel layers, and h-BN dielectric layer, as shown in **Figure R2 g**. This airgap
weakened gate modulation on the contact barrier and suppress the carrier injection from the
bottom (source) electrode, and only carrier injected from the top contact (drain) electrode. As
reported, air-gap wider than 5 nm is enough for blocking the carrier injection.^{9,10} This can be
achieved by carefully controlling the thickness of the PdSe₂ electrodes—ensuring they exceed
10 nm—and increasing the thickness of MoS₂ and WSe₂ channels. Therefore, MoS₂ and WSe₂
transistor with airgap configuration showed the unipolar n- and p-type transfer curves (**Figure**
**R2 h and i**) compared with the transistors without the airgap configuration (**Figure R2 e and**
**f**), indicating the critical role of the air-gap strategy in suppressing the ambipolar transfer
characteristics.

**Figure R2** (a) Band alignments of PdSe₂/WSe₂ and PdSe₂/MoS₂ junctions before and after contact. (b)
Potential image of Au/ PdSe₂ measured by Kelvin probe force microscopy. Scale bar: 2 μm . (c) The band
structure diagrams of PdSe₂, WSe₂, and MoS₂. (d) Schematics of top-contacted PdSe₂/WSe₂ and
PdSe₂/MoS₂, and their corresponding schematic of energy band diagram. (e, f) Transfer characteristics
of (e) PdSe₂/WSe₂, (f) PdSe₂/MoS₂ junction transistors without air-gap configuration, and corresponding
band structure diagrams inset. (g) Schematics of bottom-contacted PdSe₂/WSe₂ and PdSe₂/MoS₂, and
their corresponding schematic of energy band diagram. (h, i) Transfer characteristics of (h) PdSe₂/WSe₂,
(i) PdSe₂/MoS₂ junction transistors with air-gap configuration, and corresponding band structure
diagrams inset.

**Reference:**

- 1. Chava, P. *et al.* Electrical characterization of multi-gated WSe₂/MoS₂ van der Waals
heterojunctions. *Sci. Rep.* **14**, 5813 (2024).
2. Liu, Y., Stradins, P. & Wei, S.-H. Van der Waals metal-semiconductor junction: Weak Fermi level
pinning enables effective tuning of Schottky barrier. *Sci. Adv.* **2**, e1600069 (2016).

- 3. Nourbakhsh, A., Zubair, A., Dresselhaus, M. S. & Palacios, T. Transport Properties of a MoS₂/WSe₂
Heterojunction Transistor and Its Potential for Application. *Nano Lett.* **16**, 1359–1366 (2016).
- 4. Schulman, D. S., Arnold, A. J. & Das, S. Contact engineering for 2D materials and devices. *Chem.*
*Soc. Rev.* **47**, 3037–3058 (2018).
- 5. Zhou, Y. *et al.* Contact-engineered reconfigurable two-dimensional Schottky junction field-effect
transistor with low leakage currents. *Nat. Commun.* **14**, 4270 (2023).
- 6. Han, H. *et al.* Light-Triggered Anti-ambipolar Transistor Based on an In-Plane Lateral
Homojunction. *Nano Lett.* **24**, 8602–8608 (2024).
- 7. Zhang, G. *et al.* Reconfigurable Two-Dimensional Air-Gap Barristors. *ACS Nano* **17**, 4564–4573
(2023).
- 8. Li, C. *et al.* WSe₂/MoS₂ and MoTe₂/SnSe₂ van der Waals heterostructure transistors with different
band alignment. *Nanotechnology* **28**, 415201 (2017).
- 9. Qiu, C. *et al.* Scaling carbon nanotube complementary transistors to 5-nm gate lengths. *Science* **355**,
271–276 (2017).
- 10. Desai, S. B. *et al.* MoS₂ transistors with 1-nanometer gate lengths. *Science* **354**, 99–102 (2016).

**The corresponding discussions in the revised manuscript:**

“Kelvin probe force microscope reveal the opposite surface potential difference between
PdSe₂/WSe₂ junction and PdSe₂/MoS₂ junction as shown in **Supplementary Figure 7a and 7b**. ...
revealing the n-type and p-type contact barrier heights, with $\Phi_n = 160$ meV and $\Phi_p = 130$ meV,
respectively (**Figure 2b**).” (Page 5, line 24)

“More importantly, we optimized the device structure by incorporating the airgap configuration at
the bottom PdSe₂ contact. The airgaps are formed at the interfaces between the bottom electrode
PdSe₂, dielectric h-BN and channels MoS₂/WSe₂, as shown in **Figure 2c and Supplementary**
**Figure 9**. As reported,^{11, 29} the triangle-shaped gap significantly weaken the gate modulation and
widen the Schottky barriers’ width (SBW), therefore suppress the carrier injection through the
bottom PdSe₂ electrode into the channel.” (Page 6, line 5)

“**Figure 2. Optimization of junction transistors by using PdSe₂ contacts.** c) Schematic side
views of WSe₂-PdSe₂ and MoS₂-PdSe₂ transistors and their corresponding band diagrams at zero

drain bias (V_{DS}). The air-gaps only emerged at the bottom-PdSe₂ contact (PdSe_{2-B}), The Schottky
barrier height(SBH) are equal at top-PdSe₂ contact (PdSe_{2-T}) and PdSe_{2-B}, while the Schottky barrier
width(SBW) at PdSe_{2-B} is wider.” (Page 7, line 11)

**Comment 3:**

*“In Figure 2, the authors are suggested to provide a clearer illustration of the device structure.
Do the MoS₂ and WSe₂ touch by edge? Moreover, it would be better to have explicit
comparisons between the transport properties with and without the insertion of PdSe₂, as well
as the airgap.”*

**Response:**

We thank the reviewer’s valuable suggestion. We have included the optical image of the device
with a detailed description of device structure in the revised manuscript. As shown in **Figure**
**R3 a**, the MoS₂ and WSe₂ were not connected directly; instead, they are individually contacted
with the bottom PdSe₂ electrode. According to reviewer’s suggestion, we fabricated two-types
junction transistors without PdSe₂ bridge, including WSe₂/MoS₂ and MoS₂/WSe₂ junctions.
Their schematics and corresponding transfer curves of junction transistors are shown in **Figures**
**R3 b and c**. Both configurations exhibit unsuppressed current in the p- or n-branch and a
relatively low peak-to-valley ratio (PVR, $\sim 10^2$ at $V_D=1$ V). This behavior is attributed to carrier
injection through a narrowed barrier at the heterointerface under strong gate modulation.¹⁻³
To elucidate the role of the airgap in the transistor, we fabricated MoS₂/PdSe₂/WSe₂ junction
transistor without airgaps as shown in **Figure R3 d**. The transfer curve of the device shows
unsuppressed current on both n- and p-branch and low PVR ratio of 30 compared to the
MoS₂/PdSe₂/WSe₂ junction transistor with airgaps. These finds highlight the critical role of the
air-gap strategy in suppressing ambipolar transport and enhancing the performance of the anti-
ambipolar transistor.

1

2 **Figure R3 (a)** The optical image and side view schematic of MoS₂/PdSe₂/WSe₂ junction transistor. Scale
 3 bar: 5 μm. **(b)** Schematic and transfer curves of WSe₂/MoS₂ contact device (WSe₂ on top). **(c)** Schematic
 and transfer curves of WSe₂/MoS₂ contact device (MoS₂ on top). **(d)** Schematic and transfer curves of
 PdSe₂ top contact device.

**Reference:**

1. Chava, P. *et al.* Electrical characterization of multi-gated WSe₂/MoS₂ van der Waals heterojunctions.
 *Sci. Rep.* **14**, 5813 (2024).
 2. Doan, M.-H. *et al.* Charge Transport in MoS₂ /WSe₂ van der Waals Heterostructure with Tunable
 Inversion Layer. *ACS Nano* **11**, 3832–3840 (2017).
 3. Wu, D. *et al.* Visualization of Local Conductance in MoS₂ /WSe₂ Heterostructure Transistors. *Nano*
 *Lett.* **19**, 1976–1981 (2019).

**The corresponding discussions in the revised manuscript:**

“Compared with traditional PN junction transistors, such as the WSe₂-MoS₂ heterojunction, we
introduced PdSe₂ flake (**Supplementary Figure 2**) as a connecting bridge to optimize the symmetry
of transfer characteristics of MoS₂-WSe₂ transistors (**Supplementary Figure 3**). WSe₂ and MoS₂
channels are each in contact with bridged PdSe₂, but not contact directly. Optical image of device
exhibits in **Supplementary Figure 4**.” (Page 5, line 16)

“As shown in the inset of **Figure 2d and 2e**, electron injection from the PdSe₂ source to WSe₂ and
hole injection from the PdSe₂ source to MoS₂ are effectively blocked, therefore the off-state I_{DS} was
reduced to remarkably low levels of 10^{-5} nA. This phenomenon is further corroborated by the
ambipolar transfer curves shown in **Supplementary Figure 10**, where lower on-currents are
observed under reversed drain bias. As a result, these optimizations led to a dramatic improvement
in the I_{DS} on/off ratio, increasing by three orders of magnitude from 10^3 to 10^6 in transistors without
air gap optimization (**Supplementary Figure 11**).” (Page 6, line 14)

**Comment 4:**

*“In Supplementary Figure 10, is the airgap presented in the schematics? How does it affect the
tunneling process? In Figure 3d, why the writing and erasing speed in WSe₂ differ a bit large?
In Figure 3e, why the MoS₂ channel exhibits a slower operating speed comparing to WSe₂?
Can the authors provide calculations on the programming energy consumption and compare
with other devices with similar function?”*

**Response:**

We thank the reviewer’s valuable comments. The schematic band diagram in **Supplementary**
**Figure 10** (revised as **Supplementary Figure 18**) does not present the airgap. Instead, it
illustrates the tunneling path (dashed lines) along z-direction indicated by the dashline in
**Figure R4 a**. In our device, the tunneling path originates from the top-contacted PdSe₂,
traversing the Schottky barrier and the h-BN insulating layer to the floating graphene gate, as
shown in **Figure R4 a**. This tunneling pathway is consistent with previous studies on ultrafast
non-volatile flash memory devices based on van der Waals heterostructures.^{1,2} Therefore, the
air gap and bottom electrode do not contribute to the tunneling process.

In **Figure 3d**, the different writing and erasing speeds of nonvolatile WSe₂ transistor is

attributed to the asymmetric energy barrier along the tunneling path.³ Specifically, during the
erasing operation, the hole tunneling barrier height at graphene/h-BN interface ($\Phi_{\text{MGr-WSe}_2} =$
3.4 eV) is higher than at WSe₂/h-BN interface ($\Phi_{\text{WSe}_2\text{-MGr}} = 2.9$ eV), leading to slower erasing
speeds. Besides, during the writing process, Schottky barrier at PdSe₂/WSe₂ interface facilitates
a double-barrier modified FN tunnelling mechanism, enabling efficient carrier tunneling across
the h-BN layer and accelerating the writing operation speed (**Figure R4 b and c**).¹

In **Figure 3e**, PdSe₂-contacted MoS₂ transistor exhibits a slower operating speed compared
to PdSe₂-contacted WSe₂ transistor due to a higher Schottky barrier. As shown in **Figures R4**
**d and e**, the higher contact barrier at PdSe₂/WSe₂ interface impedes high-energy electron
injection into the graphene floating gate, reducing the electron injection efficiency and speed
particularly at low writing gate voltages (V_{GG}). As V_{GG} increases, the writing speeds of WSe₂
and MoS₂ transistor converge (revised **Supplementary Fig. 18**) because the injection process
becomes less limited by the Schottky barrier.

Besides, as suggested by the reviewer, we calculated the energy consumption of the WSe₂
memory for the program/erase operation by utilizing **Equation.3**

$$16 \quad E = V_{\text{GG}} \times I_{\text{GG}} \times t \quad \text{Eq.3}$$

where I_{GG} is the gate current and t is the pulse width. The minimum programming/ erasing
energy consumption of WSe₂ and MoS₂ memory is 19.1fJ/ 14.3 fJ by using the $|V_{\text{GG}}|$ of 38 V.
We compared the minimum energy consumption of our device with that reported in previous
studies on flash memory,⁴⁻¹⁶ as summarized in **Supplementary Table. 1**. The results indicate
that our device achieves energy consumption levels comparable to the record-low values
reported in the literature.

Figure R3 Tunneling mechanism of floating gate memory. **(a)** Schematic of tunneling paths on the device side view. **(b, d)** The schematic diagrams of WSe₂ and MoS₂ at flat band state along dashed lines in **(a)**, where the χ is the electron affinity, WF is the work function, E_g is the band gap, ϕ is the tunneling barrier, and ϕ_{SB} is the Schottky barrier. **(c, e)** The programming and erasing operations to **(c)** WSe₂, **(e)** MoS₂ floating memory.

Supplementary Table. 1 Comparison of Energy consumption per operation

Materials	Speed	Energy consumption per operation
This work	400ns/ 300ns	19.1fJ/ 14.3fJ
MoS ₂ /BN/MGr ⁴	4ms	771.2fJ
MoS ₂ /WSe ₂ /BN/MGr ⁵	500ns	9/15 aJ
NAND Flash ⁶	100μs	12nJ
MoS ₂ /HfO ₂ /Gr ⁷	100ms	18pJ
MoS ₂ /Oxide/AlScN ⁸	10μs	20pJ
MoS ₂ /Al ₂ O ₃ /HfO ₂ ⁹	200ms	5.2pJ
WSe ₂ /MoTe ₂ ¹⁰	1s	0.2pJ
MoS ₂ /HfO ₂ /Pt ¹¹	100ms	11nJ
BP/Al ₂ O ₃ ¹²	100ms	40pJ
InSe/InO _x /SiO ₂ ¹³	300ms	12pJ

WSe ₂ /BN/MGr ¹⁴	1 μs	0.6fJ
MoS ₂ /BN/MoS ₂ /GDYO/WSe ₂ ¹⁵	20 ns	10fJ
InS ₂ /BN/MGr ¹⁶	1ms	165aJ

**Reference:**

- 1. Liu, L. *et al.* Ultrafast non-volatile flash memory based on van der Waals heterostructures. *Nat.*
*Nanotechnol.* **16**, 874–881 (2021).
- 2. Yu, J. *et al.* Simultaneously ultrafast and robust two-dimensional flash memory devices based on
phase-engineered edge contacts. *Nat. Commun.* **14**, 5662 (2023).
- 3. Yu, J. *et al.* Tailoring lithium intercalation pathway in 2D van der Waals heterostructure for high-
speed edge-contacted floating-gate transistor and artificial synapses. *InfoMat* **6**, e12599 (2024).
- 4. Li, W. *et al.* Demonstration of Nonvolatile Storage and Synaptic Functions in All-Two-Dimensional
Floating-Gate Transistors Based on MoS₂ Channels. *ACS Appl. Electron. Mater.* **5**, 4354–4362
(2023).
- 5. Su, Z., Cheng, H., Sun, X., Sun, H. & Zuo, C. High-Performance Floating Gate Heterostructure
With WSe₂ -MoS₂ Diode Channel for Neural Synapse. *IEEE Electron Device Lett.* **44**, 1084–1087
(2023).
- 6. Xie, Y. Modeling, Architecture, and Applications for Emerging Memory Technologies. *IEEE Des.*
*Test Comput.* **28**, 44–51 (2011).
- 7. Bertolazzi, S., Krasnozhan, D. & Kis, A. Nonvolatile Memory Cells Based on MoS₂ /Graphene
Heterostructures. *ACS Nano* **7**, 3246–3252 (2013).
- 8. Liu, X. *et al.* Post-CMOS Compatible Aluminum Scandium Nitride/2D Channel Ferroelectric Field-
Effect-Transistor Memory. *Nano Lett.* **21**, 3753–3761 (2021).
- 9. Zhang, E. *et al.* Tunable Charge-Trap Memory Based on Few-Layer MoS₂. *ACS Nano* **9**, 612–619
(2015).
- 10. Park, S. *et al.* Nonvolatile and Neuromorphic Memory Devices Using Interfacial Traps in Two-
Dimensional WSe₂ /MoTe₂ Stack Channel. *ACS Nano* **14**, 12064–12071 (2020).
- 11. Migliato Marega, G. *et al.* Logic-in-memory based on an atomically thin semiconductor. *Nature* **587**,
72–77 (2020).

- 12. Tian, H. *et al.* A Dynamically Reconfigurable Ambipolar Black Phosphorus Memory Device. *ACS*
*Nano* **10**, 10428–10435 (2016).
- 13. Yang, F.-S. *et al.* Oxidation-boosted charge trapping in ultra-sensitive van der Waals materials for
artificial synaptic features. *Nat. Commun.* **11**, 2972 (2020).
- 14. Su, Z.-J. *et al.* Sub-femto-Joule energy consumption memory device based on van der Waals
heterostructure for in-memory computing. *Chip* **1**, 100014 (2022).
- 15. Li, Y. *et al.* Low-voltage ultrafast nonvolatile memory via direct charge injection through a threshold
resistive-switching layer. *Nat. Commun.* **13**, 4591 (2022).
- 16. Gao, C. *et al.* Touch-modulated van der Waals heterostructure with self-writing power switch for
synaptic simulation. *Nano Energy* **91**, 106659 (2022).

**The corresponding discussions in the revised manuscript:**

“Both WSe₂ and MoS₂ FGTs exhibit fast write and erase speeds, owing to the sharp and clean atomic
interface and the Schottky barrier assisted Fowler-Nordheim (F-N) tunneling mechanism at drain
region,^{48,49} as detailed in **Supplementary Figure 18.**” (page 8, line 24)

“The difference between programming and erasing speeds attributes to the different carrier injection
and tunneling barrier heights, as shown in **Supplementary Figure 18.**” (page 9, line 1)

“The ultra-fast write and erase speeds enable the device to exhibit low energy consumption of 19.1fJ
and 14.3 fJ for write and erase operations, respectively, which are highly competitive compared to
other reported memory devices, as illustrated in **Supplementary Table 1.**” (page 9, line 7)

**Comment 5:**

“In Supplementary Figure 14, the authors should explicitly describe the standard for
determining the abnormal and normal cardiovascular system. For example, why id 22 is
abnormal but id 21 is normal despite they look the same? And why id 32 is normal but not
passing through the threshold?”

**Response:**

We thank the reviewer for the valuable suggestion. We added description of the classification
standard in revised **Supplementary Note 2.** We use red and green background colors in each

subfigure to represent the ground truth classification of subjects with abnormal and normal
 cardiovascular systems, respectively. The classification standard follows clinical diagnostic
 records, which is established by the American Heart Association (AHA) and the European
 Society of Hypertension (ESH) ¹⁻³. Besides, in each subfigure, the solid lines represent the
 neuron's response curves in each subfigure, while the dashed lines indicate the neuron threshold
 determined by **Equation 3** in the manuscript. If the response curve aligns with the background
 color (green or red), it indicates that the device correctly evaluates the subject's cardiovascular
 condition. **Table R1** summarizes the RHR and SBP for subjects id-21, id-22, and id-32, along
 with the corresponding ground truth classification and the device's response. The discrepancy
 between the neuron's response and the ground truth classification highlights limitations in our
 current threshold settings.

To address the reviewer's specific question:

- 1. For subject **id-22**, the response curve exceeds the predefined threshold, correctly
 identifying it as abnormal despite the background being red.
- 2. For subjects **id-21** and **id-32**, the response curves remain within normal limits, yet the
 device fails to align with the green background, incorrectly classifying them as
 abnormal.

These cases, as shown in original **Supplementary Figure 14** (revised as **Supplementary**
 **Figure 23**), represent instances where the device did not accurately classify the subjects'
 cardiovascular conditions. Considering all cases, the overall classification accuracy of our
 device remains at 85%. While refining encoding method and the neuron threshold settings could
 further improve the classification accuracy, such optimization falls beyond the scope of this
 study.

**Table R1** Comparison of subjects' cardiovascular systems with different RHR and SBP.

Subject ID	RHR (BPM)	SBP (mmHg)	Ground Truth Classification	Device Response
id-21	78	118	Normal (Green)	Incorrect
id-22	102	120	Abnormal (Red)	Incorrect

Subject ID	RHR (BPM)	SBP (mmHg)	Ground Truth Classification	Device Response
id-32	85	115	Normal (Green)	Incorrect

**Reference:**

- 1. Unger, T. et al. 2020 International Society of Hypertension Global Hypertension Practice
Guidelines. *Hypertension* 75, 1334–1357 (2020).
- 2. Mancia, G. et al. 2023 ESH Guidelines for the management of arterial hypertension The Task
Force for the management of arterial hypertension of the European Society of Hypertension:
Endorsed by the International Society of Hypertension (ISH) and the European Renal Associat.
*J. Hypertens.* 41, (2023).
- 3. Mason, J. W. et al. Electrocardiographic reference ranges derived from 79,743 ambulatory
subjects. *J. Electrocardiol.* 40, 228-234.e8 (2007).

**The corresponding discussions in the revised manuscript:**

“The detailed encoding process and classification standard are described in **Supplementary Note**
**2.**” (Page 14, line 8)

“By comparing the responses distribution against established clinical diagnostic standards, we
further optimized threshold to achieve the highest classification accuracy beyond 85% (**Figure 5c**
**and 5f.**)” (Page 14, line 24)

“To align the SBP and RHR with the device response and allow for distinct differentiation, we utilize
an exponential coding strategy to compress the SBP range and direct linear mapping for the RHR
(as shown in **Figure. 5a**)....” (Supplenmentry file Page 3, line 16)

**Comment 6:**

“There is an incomplete sentence in the abstract line 21, “Finally, to highlight the device’s
application”. In Figure 2e, the y-axis unit is supposed to be A. Figure 2i should be 2h. The
pulse axis in Figure 4g seems mistakenly reversed comparing to Figure 4f.”

**Response:**

We thank the reviewer for the valuable suggestions. Following the comments, we carefully

rechecked the text and figures in manuscript and corrected the identified issues. Specifically,
 we have completed the unfinished sentence in the abstract (line 21), corrected the y-axis unit in
 **Figure 2e**, rectified the figure labels in **Figure 2**, and revised the pulse-axis scale in **Figure 4g**.
 Additionally, we revised the typo in the full name of “syscolic blood pressure”.

 **Figure 2. h)** Comparison of I_{DS} on/off ratio and PVR extracted from transfer curves. **i)** The PVR
 and SS of the PdSe₂-contacted anti-ambipolar transistor benchmarking the state-of-the-art 2D anti-
 ambipolar transistors. ^{13, 30-46}

 **Figure 4. f)** Spike amplitude-width plasticity of the non-monotonic neuron. **g)** Postsynaptic I_{DS}
 evolution under V_{GG} pulse train with different pulse settings. All postsynaptic I_{DS} values are
 measured at $V_{DS} = 1$ V, $V_{GG} = 0$ V. The spike trains including 30 pulses.

 **The corresponding discussions in the revised manuscript:**

“Furthermore, its nonvolatile performance can replicate biological neurons showing a
 reconfigurable monotonic and non-monotonic response by modulating the amplitude and width of
 presynaptic input. We encoded systolic blood pressure and resting heart rate data to train non-
 monotonic neurons.” (Page 1, line 19)

“**Figure 4. g)** Postsynaptic I_{DS} evolution under V_{GG} pulse train with different pulse settings. All
 postsynaptic I_{DS} values are measured at $V_{DS} = 1$ V, $V_{GG} = 0$ V. The spike trains including 30 pulses.”
 (Page 13, line 9)

1 “For instance, systolic blood pressure (SBP) and resting heart rate (RHR)—key indicators of
2 cardiovascular health...” (*Page.2, line 6*)

3

**Response to reviewers' comments**

**Response to reviewer 2#**

**General comments:**

*"In this study, an anti-ambipolar transistor, composed of a P-type WSe₂ and N-type MoS₂,*
*incorporates a PdSe₂ bridge between the two semiconductors to create an air gap structure.*
*This setup suppresses ambipolarity and prevents Fermi level pinning using PdSe₂ electrodes,*
*ensuring unipolar driveability. Additionally, a graphene floating gate is inserted at the front to*
*enable memory operations. By adjusting V_d, a transfer curve was produced, whose size can be*
*modulated and horizontally shifted through programming. Using this device, a new form of*
*spike-dependent plasticity was proposed for non-monotonic neurons. When SBP and RHR were*
*processed as the amplitude and width of input pulses, respectively, and 20 learning cycles were*
*conducted with these neurons, the system could classify normal and abnormal cases with an*
*accuracy of 85%.*

*Although this study is quite interesting in that authors implements non-monotonic neurons by*
*producing high-performance anti-ambipolar transistors using inorganic semiconductors, I feel*
*further consideration is necessary, Especially some concerns must be addressed before its*
*publication, which are noted below."*

**Response:**

We would like to thank the reviewer for the valuable comments and constructive suggestions.
In the revised manuscript, we have included introduction of the non-monotonic neuron, and
provided more detailed experiment data and discussions on characterizing the performance of
anti-ambipolar floating-gate transistor with airgap configuration. **We have responded point by**
**point to the reviewers' concerns, added revisions, and highlighted them in red in the**
**manuscript**, and hope that the revised manuscript will address all your concerns.

**The main responses are summarized as follows:**

Major comments:

- ➤ We have expanded and clarified the explanation of non-monotonic and monotonic
artificial neurons in the **Introduction** section. We have also provided background

information on cardiovascular health monitoring using systolic blood pressure (SBP)
and resting heart rate (RHR) data to better contextualize the application of our device.

➤ We have revised the color scheme of **Figure 3a** and added a detailed explanation of the
V_{GG} dual sweep measurement to enhance visual clarity and improve readability.

➤ We have added experimental data comparing the performance of floating gate
transistors (FGTs) with and without the air-gap configuration. The results demonstrate
that the air-gap structure plays a critical role in enhancing the memory window of MoS₂
and WSe₂ floating gate transistor.

➤ We have added the experimental characterization and discussion on the retention,
endurance, and speed of anti-ambipolar floating-gate transistors (AAFGTs).

➤ We have refabricated the device and remeasured the anti-ambipolar transfer curves
over a wider range of V_{DS} values, demonstrating that perfect scaling cannot be achieved
solely by adjusting V_{DS} .

➤ We have added a discussion on the peak height variation during parallel V_{th} shifting.
The WSe₂ and MoS₂ floating gate transistors exhibit synchronized V_{th} movement
during programming, resulting in minimal peak height variation.

➤ We have introduced the MIMIC dataset and included the relevant bibliographic data in
**Supplementary Table 2**.

➤ We rechecked and revised the figure and caption in the revised **Supplementary Figure**
**8**, incorporating the detailed method for Schottky barrier extraction.

Minor comments:

➤ We have thoroughly reviewed the manuscript for typographical errors and made the
corrections.

➤ We have added an introduction and clarification regarding the independent nature of
the MIMIC and PWH datasets.

**Comment 1:**

*"It is necessary to check for a typo on page 3, line 3, where "random blood pressure (SBP)"*
*might have been written incorrectly."*

**Response:**

We thank reviewer for the valuable suggestions. We have carefully rechecked the manuscript
and revised “random blood pressure” to “systolic blood pressure (SBP)” in the manuscript.

**The corresponding discussions in the revised manuscript:**

“For instance, systolic blood pressure (SBP) and resting heart rate (RHR)—key indicators of
cardiovascular health...” (Page.2, line 6)

**Comment 2:**

“In Figure 3-a's transfer curve, it would be better to choose a color combination with greater
contrast for the four types of $V_{GG\max}$ in order to more clearly demonstrate the characteristic
of increased hysteresis during double sweep.”

**Response:**

We thank the reviewer for the constructive comment. Following the suggestion, we have
redrawn the transfer curves in **Figure 3a** using high-contrast colors to more clearly demonstrate
the increased hysteresis observed with varying V_{GG} dual sweeping ranges.

**The corresponding discussions in the revised manuscript:**

“**Figure 3 a)** Transfer curves measured with V_{GG} round sweeping from negative to positive and
back to negative. $|V_{GG\max}|$ from 5 V to 20 V.” (Page.10, line 3)

**Comment 3:**

“In the caption of Supplementary Figure 12, there are two instances of “b).” A typo correction

*is needed.*”

**Response:**

We thank reviewer for the comment. We have carefully rechecked the supporting information
and revised the caption of **Supplementary Figure 20**.

**The corresponding discussions in the revised Supplementary File:**

**Supplementary Fig. 20** The I_{DS} - V_{GG} curves under different V_{DS} . (a) log scale, (b) linear scale.

(*Supplementary File, page 25, line 2*)

**Comment 4:**

*“For Supplementary Table 1, clarification is needed on whether the PWH-Elderly and PWH-*
*Young datasets are separate from the MIMIC I dataset. If they are separate, an explanation is*
*required on how these datasets were extracted (if they are datasets from previous studies,*
*references seem necessary).”*

**Response:**

We thank reviewer’s valuable suggestion. We have added these details and reference regarding
the MIMIC I and PWH datasets in the manuscript. The PWH-Elderly and PWH-Young are two
self-built datasets that are different from the public MIMIC-I dataset. The PWH datasets were
obtained from our previous work¹ at the Prince of Wales Hospital in Hong Kong, with approval
from the Joint Chinese University of Hong Kong–New Territories East Cluster Clinical
Research Ethics Committee and consent from the participants. The recruited subjects will be
categorized as PWH-Elderly if they are older than 65 years of age (n=38); conversely, subjects
younger than 35 years of age will be categorized as PWH-Young (n=23). The MIMIC dataset
is an open-access database provided by Beth Israel Deaconess Medical Center in Boston,

United States, which contains real-time clinical recordings from a diverse patient population,^{2,3}
and we randomly selected RHR and SBP data from 47 subjects in MIMIC-I as part of the
training dataset for our device.

**The corresponding discussions in the revised manuscript:**

“...the device-level abnormal threshold detection was carried out by processing real-time clinical
recordings from 47 randomly selected subjects from the publicly available MIMIC database
(Supplementary Figure 23) with diverse patient populations,^{63,64}” (Page 14, line 15)

“After training using the clinical external dataset, we evaluated the system’s reliability and stability
using internal-built PWH datasets from our previous research,⁶⁵ which included different age
groups (PWH-Elderly and PWH-Young, see Supplementary Table 2).” (Page 15, line 2)

“The MIMIC dataset is an open-access database provided by Beth Israel Deaconess Medical Center
in Boston, United States, which contains real-time clinical recordings from a diverse patient
population^{1,2}... and we randomly selected RHR and SBP data from 47 subjects in MIMIC-I as part
of the training dataset for our device.” (Supplementary File, Page 2, line 20)

**Reference:**

- 1. Qiu S, Yan B P Y, Zhao N. Stroke-volume-allocation model enabling wearable sensors for vascular
age and cardiovascular disease assessment[J]. *npj Flexible Electronics*, 2024, 8(1): 24.
- 2. Moody, G. B., Mark., R. G. A database to support development and evaluation of intelligent
intensive care monitoring. *Comput. In Cardiol.*, 657-660 (1996).
- 3. Goldberger, A. L. et al. *PhysioBank, PhysioToolkit, and PhysioNet*. *Circulation* 101, e215–e220
(2000).

**Comment 5:**

“In the introduction, the explanation of artificial neurons is quite lacking, and the explanations
of non-monotonic and monotonic neurons are also insufficient. Additionally, moving parts of
the SBP and RHR content from the Supplementary Note to the introduction or the clinical data
categorization section would aid reader comprehension. The classification of data using SBP

*and RHR for normal/abnormal conditions seems to appear too abruptly.”*

**Response:**

We thank the reviewer for the valuable suggestion. In response to your comment, we have
expanded and clarified the explanation of artificial neurons in the introduction, including the
comparison between monotonic and non-monotonic neurons. Generally, non-monotonic
neurons can selectively respond to intermediate intensity ranges, which is critical for effectively
processes data with normal distribution, such as SBP and RHR. Furthermore, we have
incorporated content from the **Supplementary Note** regarding SBP and RHR into the
introduction to provide a clear context for their relevance to cardiovascular health.

**The corresponding discussions in the revised manuscript:**

*“The transformation of monotonic stimuli from sensory inputs into non-monotonic representations*
*is fundamental to how humans process and respond to complex signals.^{1,2} ... For instance, systolic*
*blood pressure (SBP) and resting heart rate (RHR)—key indicators of cardiovascular health—often*
*fall within moderate ranges associated with healthy conditions. Effective processing of these signals*
*requires recognizing and responding to intermediate intensity ranges rather than single-side extreme*
*values, making non-monotonic encoding essential. Inspired by this biological mechanism, artificial*
*non-monotonic neurons can be implemented in health monitoring devices to preprocess*
*physiological signals, reducing the computing burden while providing direct assessments of*
*cardiovascular health.” (Page 1, line 27)*

**Comment 6:**

*“On page 7, line 7, there is no explanation of the V_{GG} double sweep, making it difficult to*
*understand. Without this explanation, Figure 3a could lead to misunderstanding, making it*
*seem like the V_{th} changes without reason.”*

**Response:**

We thank the reviewer’s constructive suggestion. We have added a detailed explanation of the
V_{GG} dual sweep measurement to improve clarity. **Figure 3a** shows the transfer curves of WSe₂
and MoS₂ FGTs under varying bidirectional V_{GG} sweeping ranges. Both counterclockwise

hysteresis memory windows of the WSe₂ FGT and clockwise hysteresis memory windows the
 MoS₂ FGT expand as the V_{GG} sweeping range increases from ± 5 V to ± 20 V. The hysteresis
 originates from the charge trapping in the FG layer, which leads to opposite shifting of the
 threshold voltage (V_{th}) during bidirectional V_{GG} sweeping. A wider V_{GG} sweeping range
 enhances the charge trapping effect, leading to larger V_{th} shifts and broader memory windows.
 In WSe₂ FGT device, a more negative V_{GG} strengthen hole accumulation in the FG, resulting in
 leftward V_{th} shifts, while positive V_{GG} facilitates hole retreat, causing rightward V_{th} shifts
 (**Figure R1 a and b**). MoS₂ FGT device exhibits analogous behaviour with electrons: a more
 positive V_{GG} enhances electron tunneling into the FG, producing rightward V_{th} shift, whereas
 negative V_{GG} promotes electron tunnelling back to the channel, leading to the leftward V_{th} shift
 (**Figure R1 c and d**). Hence, the wider V_{GG} sweeping range enhances the charge trapping effect,
 leading to larger V_{th} shifts and broader memory windows.

**Figure R1 (a, b)** Band diagrams of WSe₂ FGFET device of positive and negative global gate voltage
 (V_{GG}) operations and corresponding schematic illustration of transfer curve shifts. **(a)** Negative V_{GG} , **(b)**
 Positive V_{GG} . **(c, d)** Band diagrams of MoS₂ FGFET device of positive and negative V_{GG} operations and
 corresponding schematic illustration of transfer curve shifts. **(c)** Positive V_{GG} , **(d)** Negative V_{GG} .

**The corresponding discussions in the revised manuscript:**

**“Figure 3a** shows the transfer curves of the WSe₂ and MoS₂ floating-gate transistors (FGTs) with
 V_{GG} round sweeping ranging from ± 5 V to ± 20 V. The transfer curves exhibit the
 counterclockwise/clockwise memory window (MW), which increases linearly with the increasing
 V_{GG} sweeping range and maintains the high on/off ratio over 10^6 , as shown in **Figure 3b.**”(page 8,

line 9)

“**Figure 3. a)** Transfer curves measured with V_{GG} round sweeping from negative to positive and
back to negative. $|V_{GG}|_{max}$ from 5 V to 20 V.” (page 10, line 3)

**Comment 7:**

“On page 7, line 11, the phrase “air gaps near the bottom PdSe₂ bridge well repressed the
ambipolarity, resulting in a single counterclockwise or clockwise memory window” is used. If
there is data showing the transfer characteristics without the air gap in this device structure, it
would make it easier for readers to understand how ambipolar conversion and off-current
suppression are achieved.”

**Response:**

We thank the reviewer’s valuable suggestions. We fabricated the top-contacted WSe₂ and MoS₂
FGTs without air-gap configuration. The schematics of these devices are shown in **Figure R2**
**a**. Compared their transfer characteristics with the transistors with air-gap configuration, both
top-contacted WSe₂ and MoS₂ transistor shows the V-shaped ambipolar I_{DS} - V_{CG} curves (**Figure**
**R2 b and c**), this behavior is attributed to the carrier injection from both source and drain sides,
as depicted in the schematic band diagram in **Figure R2 d**.¹ Furthermore, for the top-contacted
FGTs, the shift of the minimum I_{DS} in the V-shaped I_{DS} - V_{CG} curves during the bi-directional V_{GG}
sweeping induced the butterfly-shaped hysteresis loops, as shown in **Figure R2 e and f**.
These butterfly-shaped hysteresis loops resulted in showed lower resistance state ratio in the
device with air gap. For example, the ratio of the high resistance state (HRS) to the low
resistance state (LRS) is only ~16 in the WSe₂ transistor without air-gap configuration.

**Figure R2 (a)** The schematics of WSe₂ and MoS₂ FET without air-gap configuration. **(b, c)** The
 comparison between I_{DS} - V_{CG} curves of **(b)** WSe₂ and **(c)** MoS₂ transistors with and without air gap
 configuration. **(d)** The band alignments of channel after different V_{GG} operations. **(e, f)** The ambipolar
 hysteresis of I_{DS} - V_{GG} curves of **(e)** WSe₂ and **(f)** MoS₂.

**The corresponding discussions in the revised manuscript:**

“Strong charge transfer typically induces pronounced ambipolarity in 2D FGT, leading to a
 reduction in the MW and on/off ratio (**Supplementary Figure 17**).⁴⁷ Here, however, the air gaps
 near the bottom PdSe₂ bridge well repressed the ambipolarity, enabling a distinct single
 counterclockwise or clockwise MW.” (Page 8, line 12)

**Reference:**

1. Zhou, Y. *et al.* Contact-engineered reconfigurable two-dimensional Schottky junction field-effect
 transistor with low leakage currents. *Nat. Commun.* **14**, 4270 (2023).

**Comment 8:**

“In Figure 3, while there is an overall explanation of the memory characteristics of P-type and
 N-type single devices, it would be reasonable to use single devices for multistate (MW)
 measurements. However, for retention, endurance, and speed, there could be significant
 performance differences between " memory devices and single semiconductor memory devices.

*Therefore, memory characteristics should be presented for AAT devices, not single*
*semiconductor devices.”*

**Response:**

We thank reviewer for the constructive suggestion. We added the characterization and
discussion on the retention, endurance, and speed of anti-ambipolar floating-gate transistor
AAFGT. For retention performance assessment, we first set the AAT curve (red line in **Figure**
**R4 a**) by sweeping from $V_{GG} = -15$ V. The output current measured at $V_{GG}=0$ V and $V_{DS}=1$ V
maintained a low current level for 2,500 s (valley state in **Figure R4 b**). Similarly, when the
AAT curve was set by sweeping from positive $V_{GG} = 10$ V (blue line in **Figure R4 a**), the high
current level read at $V_{GG}=0$ V remained for 2,500 s (peak state in **Figure R4 b**). The shorter
retention time compared to individual FGTs is primarily attributed to the less balanced charge
storage in intermediate states. Nevertheless, the stable memory states ensure reliable device
transitions.

The cyclic endurance performance of the AAFGT was characterized by repeatedly
measuring the $I_{DS}-V_{GG}$ curves with bidirectional V_{GG} sweeping as shown in **Figure R4 c**. After
110 cycles of V_{GG} sweeps, the device $I_{DS}-V_{GG}$ curves remained closely overlapped. The peak
positions V_{peak} during bi-directional V_{GG} sweeping is shown in **Figure R4 d**, confirming the
endurance of the device, although the endurance cycles were limited by the sweeping
measurement method.

To evaluate the operation speed, we first set the AAT peak near $V_{GG}=0$ V by sweeping V_{GG}
from 10 V. The sampling current measured at $V_{GG}=0$ V exceeding 1nA confirming the peak
location. Subsequently applied V_{GG} pulses induced either leftward or rightward peak shift,
resulting in decreased I_{DS} at $V_{GG}=0$ V. The AATFGT demonstrated a slightly slower speed to
compared to the separate devices, achieving successful programming with short pulses width
of 500 ns/ 1 μ s ($V_{GG}=\pm 15$ V).

**Figure R3** (a) Peak shift of AAT curves sweeping from $V_{GG} = -15$ V/ $+10$ V operations, respectively. (b)
 Retention of output current at $V_{GG} = 0$ V over 2, 500 s after operations in (a). (c) AAT curves during 110
 cyclic V_{GG} round sweep from -15 V to $+15$ V and back to -15 V. (d) the writing and erasing peaks position
 according to the 110 endurance cycles. (e, f) Set peak state as initial state, the output current after applying
 varying duration of $V_{GG} = +15$ V programming (e) and $V_{GG} = -15$ V erasing (f) pulses.

**The corresponding discussions in the revised manuscript:**

“We also evaluated the retention performance in **Figure 4c**, the peak shift after
 programming/erasing $V_{GG} = -15$ V/ $+10$ V. And the output current read at $V_{GG} = 0$ V demonstrates
 negligible degradation in both the peak (high) and valley (low) current level over 2500 s. Cyclic
 endurance of the AATFGT was characterized by repeatedly measuring the $I_{DS} - V_{GG}$ curves with
 bidirectional V_{GG} sweeping. After 110 cycles of V_{GG} sweeps, the device $I_{DS} - V_{GG}$ curves remained
 closely overlapped, as in **Supplementary Figures 22a and 22b**. The switching speeds of AATFGT

are provided in **Supplementary Figures 22c and 22d**. Compared with the individual WSe₂ and
MoS₂ FGTs, the AATFGT demonstrated a slightly slower speed, achieving successful programming
with short pulses width of 500 ns/ 1 μ s ($V_{GG} = \pm 15$ V). It is worth noting that the data in **Figure 4c**
and **Supplementary Figure 22** were obtained from a different device with the same
structure.”(Page 11, line 20)

**“Figure 4. Spike-dependent plasticity of the non-monotonic neuron c) Retention of peak and valley**
**current levels over 2, 500 s after $V_{GG} = +10$ V/ -15 V, respectively.”** (Page 13, line 4)

**“Supplementary Fig. 22 Memory performances of AATFET. (a) AAT curves during 110 cyclic V_{GG}**
**round sweep from -15 V to +15 V and back to -15 V. (b) the writing and erasing peaks position**
**according to the 110 endurance cycles. (c, d) Set peak state as initial state, the output current after**
**applying varying duration of $V_{GG} = +15$ V programming (c) and $V_{GG} = -15$ V erasing (d) pulses.”**

(Supplementary File, Page 13, line 4)

**Comment 9:**

*“On page 9, line 21, it is mentioned that the peak of the anti-ambipolar transfer curve can be*
*adjusted by controlling V_{DS} , as shown in Supplementary Figure 12, thereby increasing the*
*tunability of the neuron. However, if we refer to Supplementary Figure 12, the peak differences*
*are not significant. It seems necessary to remeasure with a larger difference in V_{DS} . Additionally,*
*when adjusting V_{DS} , not only the peak but also the threshold voltage in the P-type current region*
*changes. Therefore, it should be clearly stated that perfect scaling is not achievable by*
*adjusting V_{DS} alone.”*

**Response:**

We thank reviewer’s insightful comments. we have refabricated the device and remeasured the
anti-ambipolar transfer curves under a larger range of V_{DS} values and revised the statement in
the manuscript accordingly. The updated transfer curves with various V_{DS} conditions are shown
in **Figure R4a**. As V_{DS} increased from 0.5 V to 2.0 V, the peak current I_{peak} of anti-ambipolar
transfer curves significantly increased from ~0.6 nA to 45 nA.

As noted by the reviewer, increasing V_{DS} not only affected the I_{peak} but also resulted in
reduced symmetry and increased full width at half maximum of the anti-ambipolar transfer

curves, primarily due to a larger shift in the threshold voltage V_{th} of the p-branch transfer curve
 ($V_{th-right}$), as illustrated in **Figure R4b**. Based on these findings, we have revised the manuscript
 to clarify that although V_{DS} modulation is effective for adjusting both the height and shape of
 the transfer curves, achieving perfect symmetry scaling is not feasible with V_{DS} adjustments
 alone. Additional control approach, such as dual-gate modulation, are necessary to address this
 limitation comprehensively.¹⁻⁴

**Figure R4 (a)** The I_{DS} - V_{GG} curves under different V_{DS} . **(b)** The asymmetric threshold voltage (V_{th})
 shifts at left and right sides under different V_{DS} .

**The corresponding discussions in the revised manuscript:**

“Besides, the I_{DS} peak amplitude can be modulated by V_{DS} , as shown in **Supplementary Figure 20**.
 However, increasing V_{DS} also causes a large degradation on the symmetry of the bell-shaped curves.
 Additional control approach, such as dual-gate modulation, are necessary to facilitate further scaling
 of V_{DS} .^{12,22,23,62}” (Page 11, line 6)

**Reference:**

1. Sebastian, A., Pannone, A., Subbulakshmi Radhakrishnan, S. & Das, S. Gaussian synapses for
 probabilistic neural networks. *Nat. Commun.* **10**, 4199 (2019).
 2. Yan, X. *et al.* Reconfigurable mixed-kernel heterojunction transistors for personalized support
 vector machine classification. *Nat. Electron.* **6**, 862–869 (2023).
 3. Beck, M. E. *et al.* Spiking neurons from tunable Gaussian heterojunction transistors. *Nat. Commun.*
 **11**, 1565 (2020).
 4. Lee, C. *et al.* Highly parallel and ultra-low-power probabilistic reasoning with programmable

gaussian-like memory transistors. *Nat. Commun.* **15**, 2439 (2024).

**Comment 10:**

*“On page 9, line 25, it is stated that the transfer curve’s peak height can be maintained while*
*shifting the peak position through programming. For the peak height to remain constant, the*
*threshold voltages of the P-type and N-type semiconductors must move by the same amount*
*through programming, which is very difficult to achieve. Additionally, in Figure 4C, it appears*
*that the peak height remains constant because the graph is plotted on a log scale, but it could*
*show significant differences if plotted on a linear scale. Since maintaining a consistent peak*
*height is critical when using this curve for neurons, Figure 4C should be presented on a linear*
*scale, and data showing consistent peak height should be added.”*

**Response:**

We thank reviewer’s valuable comments and suggestions. We have provided the anti-ambipolar
transfer curves plotted in linear scale and revised our statement accordingly. Additionally, we
added a discussion on the peak height variation during the V_{th} shift. As the reviewer noted, the
peak height in linear scale exhibits a certain degree of fluctuation, as shown in **Figure R5 a**,
with a standard deviation (σ) of 7.74×10^{-11} A and coefficient of variation (CV) of 13.7%.
following the equations $\sigma = \sqrt{\sum(x-\mu)^2/N}$ and $CV = (\sigma/\mu) \times 100\%$, where x is individual peak
values, μ is mean peak value, and N is peak numbers. Accordingly, we have revised the
statement regarding maintaining a consistent peak height from the revised manuscript. However,
it is important to note that the CV of the peak height is significantly smaller than PVR of the
AAT memory, which consistently exceeds 10^3 , as shown in **Figure R5 b**. This ensures reliable
performance for our intended applications.

Furthermore, by utilizing a common graphene FG and a more uniform h-BN dielectric
layer, WSe_2 and MoS_2 FGTs achieved stable peak heights and synchronized V_{th} movement
during programming as shown in **Figure R5 c and d**. This property is anticipated to enhance
the consistency of the peak height during peak position shifting, especially when compared
with multi-gate AAT¹⁻⁴.

**Figure R5 (a)** Peak shifting after V_{GG} spikes with increasing amplitudes were applied in linear scale. **(b)**

The peak height and peak-to-valley ratio (PVR) according to the peak shift in **(a)**. **(c)** The uniform AAT

peak shifting realized by common FG structure. **(d)** The peak height variation and peak position shifts

extracted from **(c)**.

Reference:

1. Sebastian, A., Pannone, A., Subbulakshmi Radhakrishnan, S. & Das, S. Gaussian synapses for
probabilistic neural networks. *Nat. Commun.* **10**, 4199 (2019).

2. Yan, X. *et al.* Reconfigurable mixed-kernel heterojunction transistors for personalized support
vector machine classification. *Nat. Electron.* **6**, 862–869 (2023).

3. Beck, M. E. *et al.* Spiking neurons from tunable Gaussian heterojunction transistors. *Nat. Commun.*
**11**, 1565 (2020).

4. Lee, C. *et al.* Highly parallel and ultra-low-power probabilistic reasoning with programmable
gaussian-like memory transistors. *Nat. Commun.* **15**, 2439 (2024).

The corresponding discussions in the revised manuscript:

“After applying a V_{GG} spike ranging from 2 V to 10 V, the I_{DS} peak position equidistantly rightward

shifted to 4 V. This parallel V_{th} movement is mainly attributed to the common graphene floating

gate and a uniform h-BN dielectric layer. Despite the fluctuations observed among the I_{DS} peaks

**(Supplementary Figure 21)** with a standard deviation σ of 7.74×10^{-2} nA ($\sigma = \sqrt{\sum(x-\mu)^2/N}$ and,
where x is individual peak values, μ is mean peak value, and N is peak numbers.), the PVR
is consistently maintained at a value greater than 10^3 .” (Page 11, line 11)

**Comment 11:**

*“In Figure 5b, it would help readers understand why the MIMIC dataset was selected, what*
*kind of data it is, and what processing was done on the raw data. This should be specified more*
*clearly in the Supplementary Note. Additionally, bibliography data for references 6 and 7 are*
*missing from Supplementary Table 1 (these seem to be MIMIC-I references).”*

**Response:**

We appreciate the reviewer’s constructive suggestion. To address the reviewer’s concerns, we
have provided a more detailed description of the data processing steps in the revised
**Supplementary Note 2**. Additionally, references 6 and 7 have been included in
**Supplementary Table 2**. The MIMIC dataset is an open-access database provided by Beth
Israel Deaconess Medical Center in Boston, United States, which contains real-time clinical
recordings from a diverse patient population^{1,2}. It contains ECG, ABP (arterial blood pressure),
PAP (pulmonary arterial pressure), CVP (central venous pressure), and PLE (fingertip
plethysmograph) recorded signals, etc. Most importantly, this well-established dataset offers a
sufficiently large sample size, ensuring a broad range of resting heart rate (RHR) and systolic
blood pressure (SBP) values, which is critical for training and validating our device.
Furthermore, the MIMIC dataset utilizes arterial line to record continuous blood pressure that
is a gold standard method for blood pressure monitoring, and therefore, enhances data reliability.
In the data processing, we extracted the peaks from the arterial waveform which corresponding
to the value of SBP in each cardiac cycle. furthermore, within the same record of arterial
waveform, we calculated the peak-peak interval to obtain the RHR. This technique enables
accurate and synchronous extraction of both RHR and SBP.

**The corresponding discussions in the revised manuscript:**

“To be specific, the device-level abnormal threshold detection was carried out by processing real-
time clinical recordings from 47 randomly selected subjects from the publicly available MIMIC
database (**Supplementary Figure 23**) with diverse patient populations,^{63,64} as shown in **Figure 5b.**”
(*Page 14, line 15*)

“The MIMIC dataset is an open-access database provided by Beth Israel Deaconess Medical Center
in Boston, United States, which contains real-time clinical recordings from a diverse patient
population^{1,2}. ... which is critical for training and validating our device.”(*Supplementary Note 2,*
*Page 2, line 20*)

“In the data processing, we extracted the peak values from the arterial waveform from the
datasets which corresponding to the value of systolic blood pressure (SBP) in each cardiac cycle.
furthermore, within the same record of arterial waveform, ... This technique allows accurate
and synchronous extraction of both RHR and SBP.”(*Supplementary Note 2, Page 2, line 22*)

**Reference:**

- 1. G. B. Moody and R. G. Mark, “A database to support development and evaluation of intelligent
intensive care monitoring,” *Comput. Cardiol.*, pp. 657–660, Sep. 1996.
- 2. A. L. Goldberger et al., “PhysioBank, PhysioToolkit, and PhysioNet: Components of a new
research resource for complex physiologic signals,” *Circulation*, vol. 101, no. 23, pp. e215–
e220, 2000.

**Comment 12:**

“*In Supplementary Figure 4, when extracting the Schottky barrier, it is indicated that the same*
*polarity of drain voltage was used for both the electron and hole sides, but different polarities*
*should be used. This needs to be rechecked. Additionally, the equation for the current density*
*used in the thermionic emission regime for barrier height extraction should be added to the*
*explanation of Supplementary Figure 4.*”

**Response:**

We thank reviewer’s valuable suggestions. We rechecked and revised the figure and caption in
the revised **Supplementary Figure 8**. Additionally, we added the detailed method description

of Schottky barrier extraction. We utilized different drain voltage (V_{DS}) polarities for WSe₂ and
 MoS₂ transistors, reflecting the asymmetric contact properties, as discussed in the manuscript.
 Due to the screening effect of bottom PdSe₂ electrode (source) and the airgap, carriers can only
 be injected from the top PdSe₂ electrode (drain). Specifically, we selected $V_{DS} = 1V$ to facilitate
 hole injection from the drain into WSe₂, while $V_{DS} = -1V$ to was chosen to enable electron
 injection from the drain into MoS₂. The Schottky barrier is obtained from the slope of a linear
 fit to $\ln(I_{DS}/T^{1.5})$ versus $1/k_B T$, by employing the thermionic emission equation (**Equation 4**) for
 2D transport channel:

$$I_{DS} = \left[A^* T^{1.5} \exp\left(-\frac{q\Phi_B}{k_B T}\right) \right] \left[\exp\left(\frac{qV_{DS}}{k_B T} - 1\right) \right] \quad \text{Eq. 4}$$

Where I_{DS} is the current density, A^* is the effective Richardson–Boltzmann constant, T is
 temperature, q is the electron charge, Φ_B is the Schottky barrier height, and k_B is the Boltzmann
 constant.

**The corresponding discussions in the revised manuscript:**

“**Supplementary Fig. 8** Schottky barrier of PdSe₂-contacted WSe₂ and MoS₂ are calculated by 2D
 thermionic emission equation: $I_{DS} = \left[A^* T^{1.5} \exp\left(-\frac{q\Phi_B}{k_B T}\right) \right] \left[\exp\left(\frac{qV_{DS}}{k_B T} - 1\right) \right]$. Where I_{DS} is the
 saturation current density, A^* is the effective Richardson–Boltzmann constant, T is temperature,
 q is the electron charge, Φ_B is the Schottky barrier height, and k_B is the Boltzmann constant.
 The Schottky barrier height is extracted under a flat-band gate voltage condition, which was
 responsible for the start of deviations from the linear behavior. **a, b**) Schematic structure of **(a)**
 PdSe₂-WSe₂, **(b)** PdSe₂- MoS₂ contact. **c, d**) I_{DS} - V_G curves of **(c)** PdSe₂-WSe₂ transistor **(d)** PdSe₂-
 MoS₂ transistor at varying temperatures ranging from 310 K to 180 K. **e**) Arrhenius plots of
 $\ln(I_{DS}/T^{1.5})$ versus 1000/K at varying gate voltages from -1.6 V to +1.6 V of PdSe₂-WSe₂ contact. **f**)
 Arrhenius plots of $\ln(I_{DS}/T^{1.5})$ versus 1000/K at varying gate voltages from -2 V to -5.6 V of
 PdSe₂/MoS₂ contact. **g, h**) Barrier heights of the **(g)** PdSe₂-WSe₂ and **(h)** PdSe₂-MoS₂ Schottky
 junctions as a function of V_G . The Schottky barrier height is extracted under a flat-band voltage
 (V_{FB}).” (Supplementary File, page 13, line 2)

**Comment 13:**

“In the equation for setting pulse width in Supplementary Note 2, if the resting heart rate data
exceeds a certain range, it seems to go beyond the suggested manipulation width. Additional
explanation is needed for the basis of this equation. The same applies to the construction of the
SBP encoder equation below. Providing more examples and explaining why these settings were
made would help readers understand better.”

**Response:** We thank the reviewer for the valuable suggestion and correction. We have double-
checked and revised the manipulation width to the range of [0.2–1] ms, with the pulse amplitude
and the pulse width defined as:

$$10 \quad Pulse_Width (ms) = \begin{cases} 0.2 \cdot round\left(\frac{4 \cdot (RHR - 120)}{70} + 5\right), & RHR \leq 128 \\ 1, & RHR > 128 \end{cases}$$

$$12 \quad Combination_{encoder} = 2^{0.0437 \cdot SBP \pm 1.8826} \cdot \frac{Pulse_width}{0.2}, \quad 50 \leq SBP \leq 210$$

$$14 \quad Pulse_Amplitude (V) = \begin{cases} \frac{8 \cdot (CE - 12)}{10} + 13, & CE < 12 \\ \frac{18 \cdot (CE - 30)}{18} + 31, & 12 \leq CE < 30 \\ \frac{4 \cdot (CE - 450)}{420} + 35, & 30 \leq CE \end{cases}$$

The current pulse encoding model is based on MIMIC^{1,2} and PWH³ dataset. To ensure that the
constructed model adequately covers the full range of observed RHR and SBP, we conducted a
statistical analysis of the data distribution and calculated the full range of RHR and SBP in the
MIMIC and PWH datasets (see **Table R2**). Both ranges fall well within the operational limits
of our encoding model, demonstrating its applicability across typical cardiovascular states.

**Table R2** Statistics for RHR and SBP in MIMIC and PWH datasets

Datasets	Parameters	MD ± SD	95% Range	Range
MIMIC-I	SBP	119±28	[81,174]	[58, 206]
	RHR	86±17	[58,117]	[51,127]

PWH-Elderly	SBP	141±19	[110,172]	[99,180]
	RHR	70±10	[54,89]	[45,94]
PWH-Young	SBP	113±8	[103,129]	[99,143]
	RHR	73±7	[62,85]	[60,86]
Total	SBP	125±25	[89,171]	[58,206]
	RHR	78±15	[58,110]	[45,127]

**The corresponding discussions in the revised manuscript:**

“The device has an input voltage range of [0~30] V and a manipulation width of [0.2 -1] ms.

Therefore, the pulse width encoder is expressed in **Eq.2**.

$$5 \quad Pulse_Width (ms) = \begin{cases} 0.2 \cdot round\left(\frac{4 \cdot (RHR - 120)}{70} + 5\right), & RHR \leq 128 \\ 1, & RHR > 128 \end{cases} \quad (\text{Eq. 2})”$$

(*Supplementary File, Page 3, line 21*)

“The SBP encoder formulation (**Eq.3**) is developed by aligning the values $x=[70 \ 105 \ 160]$, where
these values are within the practical SBP range [50, 210] mmHg, with the corresponding SBP
encoder powers=[1 3 5],

$$10 \quad SBP_{encoder_power} = 0.0437 \cdot SBP + -1.8826, 50 \leq SBP \leq 210 \quad (\text{Eq. 3})”$$

(*Supplementary File, Page 3, line 27*)

**References:**

- 1. Moody, G. B., Mark., R. G. A database to support development and evaluation of intelligent
intensive care monitoring. *Comput. In Cardiol.*, 657-660 (1996).
- 2. Goldberger, A. L. et al. PhysioBank, PhysioToolkit, and PhysioNet. *Circulation* 101, e215–
e220 (2000).
- 3. Qiu S, Yan B P Y, Zhao N. Stroke-volume-allocation model enabling wearable sensors
for vascular age and cardiovascular disease assessment[J]. *npj Flexible Electronics*, 2024,
8(1): 24.

**Response to reviewers’ comments**

**Response to reviewer 3#**

**General comments:**

*“The authors reported a 2D anti-ambipolar transistor with a floating gate structure, which*
*functions as a non-monotonic neuron, demonstrating unsupervised learning and intermediate*
*recognition capabilities. Experimentally, the authors employed this non-volatile 2D anti-*
*ambipolar transistor to process physiological data, including blood pressure and resting heart*
*rate, achieving device-level predictions of cardiovascular risk. The concept of non-monotonic*
*response is intriguing, and the device exhibited a benchmark high PVR of ~105 and symmetric*
*n and p type non-volatile properties. However, some minor concerns need to be addressed*
*before publication”*

**Response:**

We would like to thank the reviewer for their valuable comments and constructive suggestions.

In the revised manuscript, we have expanded the background on non-monotonic neurons in the
Introduction. We also included more detailed experimental data and discussions on the
characterization of the anti-ambipolar floating-gate transistor with the airgap configuration.
Additionally, we have provided a comprehensive introduction to the design of SBP and RHR
encoders, along with the criteria for classifying normal and abnormal cardiovascular conditions.

**We have responded point by point to the reviewers’ concerns, added revisions, and**
**highlighted them in red in the manuscript**, and hope that the revised manuscript will address
all your concerns.

**The main responses are summarized as follows:**

- ➤ We have revised **Figure 1** to more intuitively illustrate the role of non-monotonic
neurons and their advantages in processing normally distributed physiological data.
- ➤ We have added an explanation of the asymmetric shifting of the threshold voltage (V_{th})
with varying V_{DS} in the anti-ambipolar transistors.
- ➤ We have expanded the discussion on the role of the airgap configuration in suppressing
ambipolar behavior in the transfer curves. We also included transfer curves of MoS₂
and WSe₂ transistors with reversed V_{DS} to indicate the effect of charge injection from

the contact barrier. Additionally, we have provided a detailed explanation of the
calculation of the Schottky barrier.

➤ We have added a detailed discussion on the parallel shifts of V_{th} . The shared graphene
floating gate and a uniform h-BN dielectric layer result in parallel V_{th} movement in
WSe₂ and MoS₂ floating gate transistors (FGTs) during the programming and erasing
process.

➤ We have added detailed introduction of design of SBP and RHR encoders in
**Supplementary Note 2.**

➤ We have added that the criteria of classification of normal and abnormal cardiovascular
system is following the clinical diagnostic records in the MIMIC and PWH datasets.

**Comment 1:**

*“In the introduction of the non-monotonic neuron, the authors should provide a clear*
*explanation of Figure 1, including the significance of the colored shaded areas, and explain*
*the advantages and necessity of using non-monotonic neurons to process SBP and RHR data.*
*This will make the concept more intuitive.”*

**Response:**

We thank the reviewer for the constructive suggestions. we have revised **Figure 1** to enhance
clarity and provided a detailed explanation of the figure, including the illustration of the colored
shaded areas. The shaded regions in **Figure 1c and 1e** represent the active response range of
the artificial neuron. Specifically, monotonic neurons have a unidirectional active response
range, defined by a single threshold, whereas non-monotonic neurons have a closed response
range with two thresholds, enabling a more flexible response to input signals.

We elaborated on the advantages and necessity of using non-monotonic neurons for
processing SBP and RHR data. Unlike monotonic neurons, which can only identify signals
above or below a threshold, non-monotonic neurons with bell-shaped response functions enable
the recognition of signals within a specific intermediate range. This capability is essential for
accurately classifying healthy cardiovascular conditions, where SBP and RHR values often fall
within these moderate ranges. Moreover, non-monotonic neurons adapt their excitatory and

inhibitory thresholds through unsupervised learning, enabling efficient and precise evaluation
 of physiological data without requiring complex programming. These features highlight the
 unique advantage of non-monotonic neurons for analyzing physiological signals like SBP and
 RHR, providing an alternative approach for the evaluation of cardiovascular risks.

 **Figure 1.** Schematic comparison between monotonic and non-monotonic neurons. **a)** Distribution
 of systolic blood pressure (SBP) and resting heart rate (RHR) and their correlation with
 cardiovascular health levels. **b)** Illustration of SBP and RHR data collection and monotonic pulse
 coding. **c)** Schematic of a monotonic excitatory neuron with a S-shaped response curve and a
 shiftable threshold voltage (V_{th}). **d)** The unidirectional active response range of a monotonic
 excitatory neuron, lack of selective recognition to intermediate input ranges. **e)** Schematic of a non-
 monotonic neuron with the bell-shaped response curve and a closed response range defined by two
 parallel shiftable V_{th} values. The shaded regions in represent the active response range of the
 artificial neuron. **f)** The closed active response range of a non-monotonic excitatory neuron, showing
 selectively response to a specific intermediate input range. The shaded regions in represent the
 active response range of the artificial neuron.

**The corresponding discussions in the revised manuscript:**

“ However, the *S*-shaped response curve poses a challenge for selectively activating signals within

an intermediate intensity range, e.g, SBP and RHR. Moderate ranges of SBP and RHR are widely
recognized as indicative of healthy cardiovascular function. ... However, using monotonic neurons
with S-shaped response function to process these signals results in recognizing only extreme
values—those exceeding or falling below the threshold—while failing to identify signals within the
moderate range, as shown in **Figure 1c and 1d**.... Additionally, these neurons adapt excitatory and
inhibitory thresholds through unsupervised learning, avoiding activation range overlap while
enabling memory formation and intensity recognition of SBP and RHR signals. (**Figure 1f**).” (Page
3, line 22)

“Non-monotonic neurons, also known as intensity-tuned neurons,³ overcome this limitation by
selectively responding to intermediate stimulus ranges.⁴ ... Effective processing of these signals
requires recognizing and responding to intermediate intensity ranges rather than single-side extreme
values, making non-monotonic encoding essential.” (Page 2, line 3)

“**Figure 1**. Schematic comparison between monotonic and non-monotonic neurons. **a)**
Distribution of systolic blood pressure (SBP) and resting heart rate (RHR) and their correlation with
cardiovascular health levels. ... **f)** The closed active response range of a non-monotonic excitatory
neuron, showing selectively response to a specific intermediate input range. The shaded regions in
represent the active response range of the artificial neuron.” (Page 4, line 18)

**Comment 2:**

“In Figure 1d, the V_{th} of both WSe_2 and MoS_2 transistors could be shifted by V_{DS} . However,
in Figure 1e, the V_{th} at left side remains unchanged. It is essential to explain the difference in
the V_{th} shift in anti-ambipolar transistor with V_{DS} .”

**Response:**

We thank reviewer for the insightful comments. We added explanation of the asymmetric
shifting of the threshold voltage (V_{th}) with various V_{DS} . As shown in the **Figure R2 a**, the V_{th} at
left side is primarily influenced by the n-type MoS_2 transistor and V_{th} at right side is primarily
influenced by p-type WSe_2 transistor. The asymmetric shifting of V_{th} in the anti-ambipolar
transistor originated from its circuit connections (**Figure R2 a**). Specifically, when the V_{DS} is
applied on the WSe_2 transistor while the MoS_2 transistor is grounded, the drain electric fields

more effectively thinning the contact barrier of the WSe₂ transistor, thereby compromising gate
 control over this transistor, and causing a rightward shift of V_{th-P} , while the V_{th-N} remains
 unaffected due to its grounded connection, as demonstrated in **Figure R2 b**. To further verify
 this mechanism, we applied opposite V_S to the MoS₂ transistor and grounded WSe₂ transistor,
 a consistent V_{th-WSe_2} and leftward shift V_{th-MoS_2} is observed in **Figure R2 c**.

 **Figure R2 (a)** The circuit diagram and schematics of formation of AAT transfer characteristic. **(b)** The
 AAT FET under varying positive drain voltage (V_D) from 0.6 ~ 1.2 V. **(c)** The AAT FET under varying
 negative source voltage(V_S) from -0.6 ~ -1.2 V.

**The corresponding discussions in the revised manuscript:**

“The rightward shift of bell-shaped peak is attributed to the stronger electric field distribution over
 the drain region compared to the source region(Supplementary Figure 15).” (Page 6, line26)

**Comment 3:**

“On page 5 line 15, the text mentioned that the air-gap occurs at the bottom PdSe₂ electrode
 contributes to optimizing the transfer characteristics. The schematic should highlight the air-
 gap and provide more detailed discussion on how this configuration influence the device
 performance. Including transfer curves with reversed V_{DS} is crucial. Additionally, the authors
 should clearly explain the methodology used to calculate the Schottky barrier.”

**Response:**

We thank reviewer for the valuable suggestions. We highlighted the air-gap configuration in the
 schematics and added discussion about the influence of air-gap configuration on the device
 performance. Additionally, we added detailed explanation on the Schottky barrier calculation.

As shown in **Figure R3 a**, the triangular air gap is formed between the bottom PdSe₂ electrodes
 and the MoS₂ and WSe₂ layers. This air gap serves to suppress ambipolar transfer characteristics
 by widening the contact barrier (SBW), thus reducing carrier injection from the source electrode.
 Specifically, the gate modulation is weakened at the contact barrier, preventing carrier injection
 from the bottom electrode (Source), while enabling carrier injection only from the top contact
 (drain).^{5,9} As a result, the MoS₂ and WSe₂ transistors with the air-gap configuration exhibit
 unipolar n-type and p-type transfer characteristics, contrasting with the ambipolar behavior
 observed in devices without the air-gap configuration (**Figure R3 b and c**). This demonstrates
 the critical role of the air-gap strategy in optimizing the transfer characteristics.

In response to the reviewer's comment on transfer curves of WSe₂ (MoS₂) with reversed
 V_{DS} , we have included these in **Figure R3 d and e**. The WSe₂ transistor at $V_{DS} = 1$ V and the
 MoS₂ transistor at $V_{DS} = -1$ V shows the unipolar p-type and n-type transfer characteristics,
 respectively. Conversely, when V_{DS} is reversed (WSe₂ at $V_{DS} = -1$ V, MoS₂ at $V_{DS} = -1$ V), both
 transistors exhibit an ambipolar trend, resulting in a reduced I_{DS} on/off ratio.

Regarding the Schottky barrier height (SBH), it is extracted using the 2D thermionic emission
 equation in conjunction with temperature-dependent measurements. According to the thermionic
 theory, when the back gate voltage was below flat-band voltage, the current can be expressed as:

$$18 \quad I_{DS} = \left[A^* T^{1.5} \exp\left(-\frac{q\phi_B}{k_B T}\right) \right] \left[\exp\left(\frac{qV_{DS}}{k_B T} - 1\right) \right]$$

where A^* is the 2D equivalent Richardson constant, S is the contact area of the junction, q is the
 elementary charge, ϕ_B is the barrier height, n is the ideality factor, and k_B is the Boltzmann constant. The
 $\ln(I_{DS}/T^{1.5})$ versus $1/k_B T$ curves at various V_{GS} are plotted in **Figure R3 f and g**. From the slope of the
 linear fit, ϕ_B was calculated and given as a function of V_{GS} in revised **Supplementary Figure 8**.

**Figure R3 (a)** Schematics of bottom-contact and top-contacted PdSe₂/WSe₂, PdSe₂/MoS₂, and the
 corresponding schematics of energy-band diagrams. SBH and SBW represent the Schottky barrier height
 and width, respectively. **(b)** The schematics and transfer curves of PdSe₂/WSe₂ FET with and without
 air-gap. **(c)** The schematics and transfer curves of PdSe₂/MoS₂ FET with and without air-gap. **(d, e)**
 Transfer curves of **(d)** PdSe₂/MoS₂ FET, **(e)** PdSe₂/WSe₂ FET with air-gap at $V_D = \pm 1$ V. **(f, g)** Arrhenius
 plots of $\ln(I_{DS}/T^{1.5})$ versus $1000/K$ at varying gate voltages of **(f)** PdSe₂/MoS₂ contact, **(g)** PdSe₂/WSe₂
 contact.

**The corresponding discussions in the revised manuscript:**

“More importantly, we optimized the device structure by incorporating the airgap into the gap at the
 bottom PdSe₂ contact, the airgaps configuration are shown in **Figure 2c** and **Supplementary**
 **Figure 9**. The triangle-shaped gap defined by the bottom electrode PdSe₂, dielectric h-BN and
 channels MoS₂/WSe₂ significantly weaken the gate modulation and widen the Schottky barriers’
 width (SBW), therefore suppress the carrier injection through the bottom PdSe₂ electrode into the
 channel. We investigated the transfer characteristics of PdSe₂ contacted transistors with airgaps
 configurations, in which top PdSe₂ serve as the drain electrodes.” (Page 6, line 7)

“As shown in the inset of **Figure 2d** and **2e**, electron injection from the PdSe₂ source to WSe₂ and
 hole injection from the PdSe₂ source to MoS₂ are effectively blocked, therefore the off-state I_{DS}
 were reduced to remarkably low levels of 10^{-5} nA. With reversed V_{ds} setting, the transistor showed
 a lower on-currents due to lower carrier injection efficiency (**Supplementary Figure 10**).
 Compared with transistors without airgap optimization, the I_{DS} on/off ratio increased by three orders
 of magnitude from 10^3 to 10^6 in transistors without airgap optimization (**Supplementary Figure**

**11).**” (Page 6, line 14)

“We also extracted the Schottky barrier height (SBH) at the interfaces of PdSe₂/MoS₂ and
PdSe₂/WSe₂ from low-temperature measurement (detailed calculation is introduced in
**Supplementary Figure 8).**” (Page 5, line 29)

**Comment 4:**

*“The anti-ambipolar transistors in previous report require the coordination of drain and dual-
gate voltages to achieve parallel shifts of the IDS peak position and maintained peak height
[Nat Commun 11, 1565 (2020)]. The authors should discuss why, in floating-gate-based
transistors, parallel shifts of the peak can be achieved using a single gate.”*

**Response:**

We thank reviewer’s valuable comment and suggestions. We have added detailed discussion
on the parallel shifts of V_{th} in revised manuscript. The programming and erasing process based
on tunneling process are schematically illustrated in **Figure R4 a**. During these processes,
carriers trapped in the floating gate induce an electrostatic doping effect, leading to a shift in
V_{th} . This behavior follows the equation:

$$\Delta V_{th} = -\frac{dQ}{\epsilon_{BN}} = -\frac{Q}{C_{BN}}$$

Where the d is the dielectric layer thickness, Q is the stored charge in the floating gate, ϵ_{BN} is
the dielectric constant of BN, and C_{BN} is the dielectric layer capacitance. Hence, by utilizing
a shared graphene floating gate and a uniform h-BN dielectric layer, WSe₂ and MoS₂ floating
gate transistor should exhibit parallel V_{th} movement after programming and erasing process.
This property is challenging to achieve in dual-gate anti-ambipolar transistors¹ due to the more
complicated device structures. To demonstrate this mechanism, we measured the shift of V_{th} of
WSe₂ and MoS₂ floating gate transistor after programmed by negative V_{GG} pulses (**Figure R4
b and c**). After applying the same V_{GG} pulse, the transfer curves of both WSe₂ and MoS₂
transistor shifted progressively leftwards. The extracted V_{th} values (**Figure R4 d**) confirm
symmetric and linear shifts, with a fixed interval of approximately 7 V between the two devices.

**Figure R4 (a)** Tunneling schematics of WSe₂ and MoS₂ devices under programming and erasing process.

**(b,c)** The transfer curves of MoS₂ **(b)** and WSe₂ **(c)** sweeping from different V_{GG} (-7 V to +3 V, step=2
4 V). **(d)** The changes of V_{th} in **(b)** and **(c)**.

**The corresponding discussions in the revised manuscript:**

“After applying a V_{GG} spike ranging from 2 V to 10 V, the I_{DS} peak position shifted equidistantly
rightward to 4 V (**Supplementary Figure 21**). This parallel V_{th} movement is mainly attributed to
the common graphene floating gate and a uniform h-BN dielectric layer. Despite the fluctuations
observed among the I_{DS} peaks (**Supplementary Figure 21**) with a standard deviation of 7.74×10^{-2}
nA ($\sigma = \sqrt{\sum(x-\mu)^2/N}$ and, where x is individual peak values, μ is mean peak value, and N is
peak numbers.), the PVR is consistently maintained at a value greater than 10^3 .” (Page 11, line 11)

**Reference:**

1. Beck, M. E. *et al.* Spiking neurons from tunable Gaussian heterojunction transistors. *Nat. Commun.*
**11**, 1565 (2020).

**Comment 5:**

“In Figure 5a and Supporting Note 2, different encoders are employed for SBP and RHR data,
respectively. it is necessary to give a more detailed explanation to the choose of encoding

*models.”*

**Response:** Thanks for the constructive suggestion. We have added additional ideas of design
of SBP and RHR encoders in **Supporting Note 2**. The SBP and RHR encoders are uniquely
designed based on the specific ranges of SBP and RHR within the datasets as outlined in **Table**
**R2**. The SBP encompasses a broad dynamic range of approximately 82 mmHg, while the RHR
is characterized by a more limited range of 52 BPM. To align the SBP and RHR with the device
response and allow for distinct differentiation, we utilize an exponential coding strategy to
compress the SBP range and direct linear mapping for the RHR (as shown in **Figure. 5a**). These
encoders are easy to implement and commonly used,^{1,2} and their combination works effectively
with the device characteristics to achieve the feature coding function.

**The corresponding discussions in the revised manuscript:**

“To align the SBP and RHR with the device response and allow for distinct differentiation, we
utilize an exponential coding strategy to compress the SBP range and direct linear mapping for
the RHR (as shown in **Fig. 5a**). These encoders are easy to implement and commonly used,^{6,7}
and their combination works effectively with the device characteristics to achieve the feature
coding function...” (*Supporting Note 2, Page 3 Line 5*)

**Reference:**

- 1. Zill D G. Advanced engineering mathematics[M]. *Jones & Bartlett Learning*, 2020.
2. Dehaene S, Izard V, Spelke E, et al. Log or linear? Distinct intuitions of the number scale in
Western and Amazonian indigene cultures[J]. *Science*, 2008, 320(5880): 1217-1220.

**Comment 6:**

“For the application, the authors should clearly introduce the criteria used to distinguish
between healthy individuals and those at risk, along with the corresponding references.”

**Response:**

Thanks for the valuable suggestions. We have added additional details of the criteria for
accuracy evaluation of abnormal cardiovascular system in the revised manuscript. The overall

accuracy of its assessment was validated against clinical diagnostic results. The clinical
standard ranges for SBP and RHR are referenced from the guidelines established by the
American Heart Association and the European Society of Hypertension¹⁻³. According to these
guidelines, the normal range for resting heart rate is 60-100 beats per minute (BPM), and the
normal range for blood pressure is 90-120 millimeters of mercury (mmHg).

**The corresponding discussions in the revised manuscript:**

“By comparing the responses distribution against established clinical diagnostic standards, we
further optimized threshold to achieve the highest classification accuracy beyond 85% (**Figure 5c**
**and 5f**.” (Page 14, Line 24)

“The clinical standard ranges for SBP and RHR are referenced from the guidelines established
by the American Heart Association and the European Society of Hypertension²⁵⁻²⁷. According
to these guidelines, the normal range for resting heart rate is 60-100 beats per minute (BPM),
and the normal range for blood pressure is 90-120 millimeters of mercury (mmHg).” (Page 18,
Line 14)

**References:**

- 1. Unger, T. *et al.* 2020 International Society of Hypertension Global Hypertension Practice
Guidelines. *Hypertension* **75**, 1334–1357 (2020).
- 2. Mancia, G. *et al.* 2023 ESH Guidelines for the management of arterial hypertension The Task
Force for the management of arterial hypertension of the European Society of Hypertension:
Endorsed by the International Society of Hypertension (ISH) and the European Renal Associat.
*J. Hypertens.* **41**, (2023).
- 3. Mason, J. W. *et al.* Electrocardiographic reference ranges derived from 79,743 ambulatory
subjects. *J. Electrocardiol.* **40**, 228-234.e8 (2007).

Response to reviewers' comments

Response to reviewer 1#

General comments:

“The authors have addressed all my questions and comments. I would like to recommend the publication in Nature communications.

Response:

We sincerely thank the reviewer for their valuable time and comments.

Response to reviewer 2#

Comment 1:

“In the revision, we requested experiments on memory characteristics in anti-ambipolar devices rather than p- and n-type unit devices. While the retention characteristics of the original unit devices were shown for 10,000 seconds, the revised file only shows 2,500 seconds for the anti-ambipolar device. It would be better if a plot extrapolated to a longer time frame could be provided.”

Response:

We thank reviewer’s valuable suggestion. We have remeasured the retention characteristic in anti-ambipolar device up to 10,000 s and extrapolated the time frame in **Figure 4c**.

The corresponding discussions in the revised manuscript:

*“We also evaluated the retention performance by monitoring the output current at $V_{GG} = 0$ V after programming/erasing operations ($V_{GG} = -15$ V / $+10$ V). As shown in **Figure 4c**, the peak position shift results in two distinct states, peak (high) and valley (low) output states. Both states demonstrate negligible degradation in both the over 10^4 s.”* (Page 8, line 25)

“Figure 4. c) Retention properties of peak and valley current states over 10,000 s after $V_{GG} = +10$ V / -15 V, respectively.” (Page 24, line 4)

Comment 2:

“The section on Schottky barrier width (SBW) has been revised, but the concept of SBW has not been clearly defined. It is unclear what specific part of the semiconductor at the contact is being referred to as SBW. Additionally, the references cited in this section do not explicitly define Schottky

barrier width. Providing a more detailed mathematical or conceptual explanation of SBW in the manuscript would help readers understand this concept more clearly.”

Response:

We thank reviewer’s valuable comments. We have added conceptual explanation of the Schottky barrier width (SBW) in the revised manuscript. SBW refers to the width of band bending region formed at the metal-2D semiconductor interface, from the metal contact edge to the point where the bands reach their flat-band position in the 2D channel area. It’s corresponding part in channel is exhibited in **Figure R1**.

Figure R1 Device schematic and band structure diagrams of top-contact and bottom-contact configurations. The gray, light blue, and dark blue regions refer to the contact edge, band bending region and intrinsic channel.

The corresponding discussions in the revised manuscript:

“More importantly, in addition to modulating the SBH, we engineered the Schottky barrier width (SBW), which refers to the lateral band bending region in channel formed at the contact area.¹¹ The corresponding device structure is optimized by incorporating the airgap configuration at the bottom PdSe₂ contact.” (Page 5, line 13)

Response to reviewer3#

General comments:

“the authors had addressed all the concern.”

Response:

We sincerely thank the reviewer for their valuable time and positive comments.